# The landscape of human mutually exclusive splicing

Klas Hatje[1,2,†], Raza-Ur Rahman[2,3], Ramon O Vidal[2,‡], Dominic Simm[1,4], Björn Hammesfahr[1,§], Vikas Bansal[2,3] (ID), Ashish Rajput[2,3] (ID), Michel Edwar Mickael[2,3], Ting Sun[2,3], Stefan Bonn[2,3,5,*] (ID) & Martin Kollmar[1,**] (ID)

## Abstract

Mutually exclusive splicing of exons is a mechanism of functional gene and protein diversification with pivotal roles in organismal development and diseases such as Timothy syndrome, cardiomyopathy and cancer in humans. In order to obtain a first genomewide estimate of the extent and biological role of mutually exclusive splicing in humans, we predicted and subsequently validated mutually exclusive exons (MXEs) using 515 publically available RNA-Seq datasets. Here, we provide evidence for the expression of over 855 MXEs, 42% of which represent novel exons, increasing the annotated human mutually exclusive exome more than fivefold. The data provide strong evidence for the existence of large and multi-cluster MXEs in higher vertebrates and offer new insights into MXE evolution. More than 82% of the MXE clusters are conserved in mammals, and five clusters have homologous clusters in *Drosophila*. Finally, MXEs are significantly enriched in pathogenic mutations and their spatio-temporal expression might predict human disease pathology.

**Keywords** alternative splicing; differential expression; mutually exclusive splicing; splicing mechanisms
**Subject Categories** Chromatin, Epigenetics, Genomics & Functional Genomics; Genome-Scale & Integrative Biology; Transcription
**Mol Syst Biol.** (2017) 13: 959

## Introduction

Alternative splicing of pre-messenger RNAs is a mechanism common to almost all eukaryotes to generate a plethora of protein variants out of a limited number of genes (Matlin *et al*, 2005; Nilsen & Graveley, 2010; Lee & Rio, 2015). High-throughput studies suggested that not only 95–100% of all multi-exon genes in human are affected (Pan *et al*, 2008; Wang *et al*, 2008; Gerstein *et al*, 2014) but also that alternative splicing patterns strongly diverged between vertebrate lineages implying a pronounced role in the evolution of phenotypic complexity (Barbosa-Morais *et al*, 2012; Merkin *et al*, 2012). Five types of alternative splicing have been identified to contribute to most mRNA isoforms, which are differential exon inclusion (exon skipping), intron retention, alternative 5′ and 3′ exon splicing, and mutually exclusive splicing (Blencowe, 2006; Pan *et al*, 2008; Wang *et al*, 2008; Nilsen & Graveley, 2010). Mutually exclusive splicing generates alternative isoforms by retaining only one exon of a cluster of neighbouring internal exons in the mature transcript and is a sophisticated way to modulate protein function (Letunic *et al*, 2002; Meijers *et al*, 2007; Pohl *et al*, 2013; Tress *et al*, 2017a). The most extreme cases known so far are the arthropod *DSCAM* genes, for which up to 99 mutually exclusive exons (MXEs) spread into four clusters were identified (Schmucker *et al*, 2000; Lee *et al*, 2010; Pillmann *et al*, 2011).

Opposed to arthropods, current evidence suggests that vertebrate MXEs only occur in pairs (Matlin *et al*, 2005; Gerstein *et al*, 2014; Abascal *et al*, 2015a), and genomewide estimates in human range from 118 (Suyama, 2013) to at most 167 cases (Wang *et al*, 2008). Despite these relatively few reported cases, mutually exclusive splicing might be far more frequent in humans than currently anticipated, as has been recently revealed in the model organism *Drosophila melanogaster* (Hatje & Kollmar, 2013). Apart from their low number, MXEs have been described in many crucial and essential human genes such as in the α-subunits of six of the 10 voltage-gated sodium channels (*SCN* genes) (Copley, 2004), in each of the glutamate receptor subunits 1–4 (*GluR1-4*) where the MXEs are called flip and flop (Sommer *et al*, 1990), and in *SNAP-25* as part of

1 Group Systems Biology of Motor Proteins, Department of NMR-Based Structural Biology, Max-Planck-Institute for Biophysical Chemistry, Göttingen, Germany
2 Group of Computational Systems Biology, German Center for Neurodegenerative Diseases, Göttingen, Germany
3 Center for Molecular Neurobiology, Institute of Medical Systems Biology, University Clinic Hamburg-Eppendorf, Hamburg, Germany
4 Theoretical Computer Science and Algorithmic Methods, Institute of Computer Science, Georg-August-University, Göttingen, Germany
5 German Center for Neurodegenerative Diseases, Tübingen, Germany
*Corresponding author. Tel: +49 40 7410 55082; E-mail: sbonn@uke.de
**Corresponding author. Tel: +49 551 5036960; E-mail: mako@nmr.mpibpc.mpg.de
†Present address: Roche Pharmaceutical Research and Early Development, Pharmaceutical Sciences, Roche Innovation Center Basel, Basel, Switzerland
‡Present address: Max-Delbrück-Center for Molecular Medicine, Berlin, Germany
§Present address: Research and Development—Data Management (RD-DM), KWS SAAT SE, Einbeck, Germany

the neuroexocytosis machinery (Johansson *et al*, 2008). Although MXEs within a cluster often share high similarity at the sequence level, they are usually not functionally redundant, as their inclusion in the mRNAs is tightly regulated. Thus, mutations in MXEs have been shown to cause diseases such as Timothy syndrome (missense mutation in the *CACNA1C* gene) (Splawski *et al*, 2004, 2005), cardiomyopathy (defect of the mitochondrial phosphate carrier *SLC25A3*) (Mayr *et al*, 2011) or cancer (mutations in, e.g., the pyruvate kinase *PKM* and the zinc transporter *SLC39A14*) (David *et al*, 2010).

Despite the implications of mutually exclusive splicing in organismal development and disease, current knowledge on the magnitude of MXE usage and its relevance in biological processes is far from complete. In order to obtain a genomewide, unbiased estimate of the extent and biological role of mutually exclusive splicing in humans, a set of 6,541 MXE candidates was compiled from annotated and novel predicted exons, and rigorously validated using over 15 billion reads from 515 RNA-Seq datasets.

## Results

### The human genome contains 855 high-confidence MXEs

Compared to other splicing mechanisms, mutually exclusive splicing in humans seems to be a rare event. MXEs are characterized by genomic vicinity, splice-site compatibility and mutually exclusive presence in protein isoforms. Accordingly, the human genome annotation (GenBank v. 37.3) contains only 158 MXEs in 79 protein-coding genes (Appendix Figs S1–S3). MXEs are often phrased "homologous exons" in the literature because they likely originated from the same ancestral exon. We refrain from using this term throughout our analysis, because several MXEs present in the genome annotation do not show any sequence homology and many neighbouring exons with high sequence similarity are not spliced in a mutually exclusive manner.

In a first attempt to chart an atlas of genomewide mutually exclusive splicing in humans, we decided to predict potential MXE candidates and validate those using published RNA-Seq data. In a first step, we generated a set of MXE candidates in the human genome

(v. 37.3) from all annotated protein-coding exons and from novel exons predicted in intronic regions including only internal exons in the candidate list (Fig 1A, Appendix Figs S1–S4). From the annotated exons, we selected those that appeared mutually exclusive in transcripts, and neighbouring exons that show sequence similarity and are translated in the same reading frame. To generate novel exon candidates, we predicted exonic regions in neighbouring introns of annotated exons based on sequence similarity and similar lengths (Pillmann *et al*, 2011). We did not consider potential MXEs containing in-frame stop codons such as the neonatal-specific MXE reported for the sodium channel *SCN8A* (Zubović *et al*, 2012), and exons overlapping annotated terminal exons (Appendix Fig S2). The reconstruction resulted in a set of 6,541 MXE candidates in 1,542 protein-coding genes, including 1,058 (68.6%) genes for which we predicted 1,722 completely novel exons in previously intronic regions (Fig 1B). Most introns in human genes are extremely long necessitating careful and strict validation of the MXE candidates to exclude false-positive predictions (Lee & Rio, 2015).

To validate the predicted MXE candidates, we made use of over 15 billion publically available RNA-Seq reads, selecting 515 samples comprising 31 tissues and organs, 12 cell lines and seven developmental stages (Barbosa-Morais *et al*, 2012; Djebali *et al*, 2012; Tilgner *et al*, 2012; Xue *et al*, 2013; Yan *et al*, 2013; Fagerberg *et al*, 2014; Dataset EV1). The data were chosen to encompass common and rare potential splice events in a broad range of tissues, cell types and embryonic stages. Accordingly, the transcription of 6,466 (99%) of the MXE candidates is supported by RNA-Seq reads mapped to the genome (Appendix Fig S3A). To be validated as true mutually exclusive splicing event, each MXE of a cluster needed to exhibit splice junction (SJ) reads from every MXE to up- or downstream gene regions bridging the other MXE(s) of the cluster (Fig 1A). In addition, MXEs should not exhibit any SJ reads to another MXE except when the combined inclusion causes a frame shift and therefore a premature stop codon (Fig 1A, Appendix Figs S3A and D, S5, and S6). These stringent criteria define a high-confidence set of MXEs, requiring three constraints for a cluster of two MXEs and already 18 constraints for a cluster of five MXEs (Appendix Fig S7). In case of clusters with more than two MXE candidates, the validation criteria were applied to the cluster including all MXE candidates as well as to all possible sub-clusters to

---

**Figure 1. The human genome contains 1,399 high-confidence MXEs.**

A   Schematic representation of the various annotated and predicted exon types included in the MXE candidate list. For MXE validation, at least three restraints must be fulfilled: the absence of an MXE-joining read (R1), except for those leading to frame shift, and the presence of two MXE-bridging SJ reads (R2 and R3).

B   Prediction and validation of 1,399 1SJ (855 3SJ) human MXEs. Top: Dataset of 6,541 MXE candidates from annotated and predicted exons. Bottom left: MXE candidates for which splice junction data are currently missing hindering their annotation as MXE or other splice variant. Bottom right: Validation of the MXE candidates using over 15 billion RNA-Seq reads. The outer circles represent the validation based on at least a single read for each of the validation criteria (1SJ), while the validation shown in the inner circles required at least three reads (3SJ).

C   MXE saturation analysis. Whereas increasing amounts of RNA-Seq reads should lead to the confirmation of further MXE candidates, more RNA-Seq reads might also result in the rejection of previously validated MXEs. The green curves show the number of validated MXEs in relation to the percentage of total RNA-Seq reads used for validation. The orange curves indicate the number of initially "validated MXEs" that were rejected with increasing amounts of reads. Grey dashed lines indicate the point of saturation, which is defined as the point where a twofold increase in reads leads to rejection of less than 1% of the validated MXEs. Of note, whereas the rejection of validated MXEs saturates with 20% of the data, the amount of novel MXE validations is still rapidly increasing.

D   Distribution of validated MXEs in two-exon and multi-exon clusters.

E   Size and distribution of multi-cluster MXEs.

F   The *CUX1* gene (cut-like homeobox 1) contains two interleaved clusters of MXEs (clusters 1 and 2) and two standard clusters each with two MXEs (clusters 3 and 4). The exon 3 and exon 4 variants each are orthologous exons. The exon 4 variants are mutually exclusive (cluster 2). Exon 3a is a differentially included exon and only spliced together with exon 4a. The exons 3b, 3c, 3d and 3e are part of a cluster of four MXEs (cluster 1) and are only spliced together with exon 4b (Appendix Figs S16 and S17). Novel exons are labelled with an asterisk.

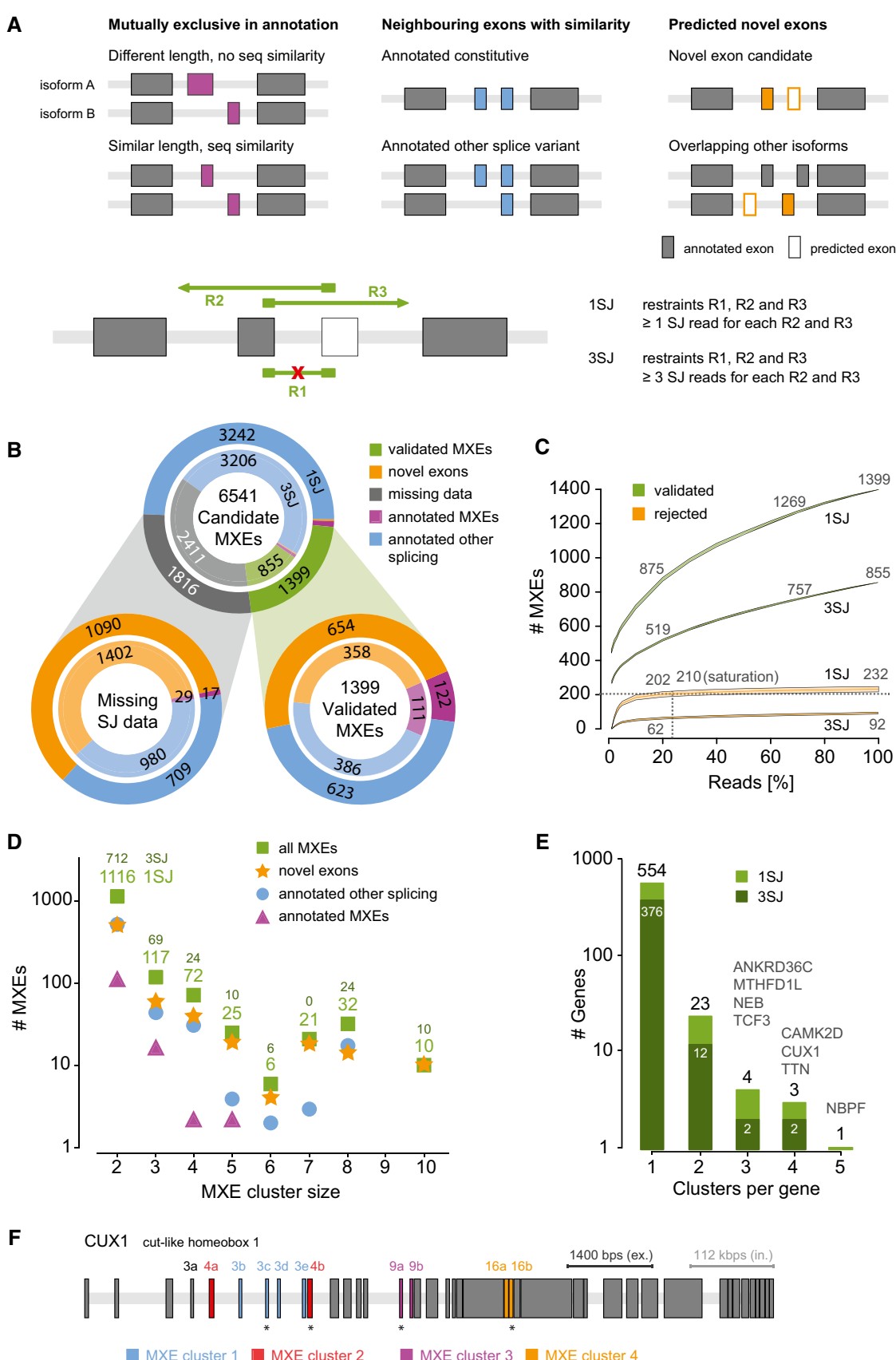

Figure 1.

identify the largest cluster fulfilling all MXE criteria. According to these criteria, 1,399 MXEs were verified with at least one SJ read per exon (1SJ), supported by 2.2 million exon mapping and 34 million SJ reads, increasing the total count of human MXEs by almost an order of magnitude (158–1,399) (Fig 1B, Dataset EV2); 855 MXEs were found to be supported by at least three splice junction reads per exon (3SJ) validated by 1.5 million exon mapping and 27 million SJ reads (Appendix Figs S3B and C, S8–S10). The 1,399 (855, numbers in brackets refer to the 3SJ validation) verified MXEs include 122 (112) annotated MXEs (Fig 1B "annotated MXE"), 623 (388) exons that were previously annotated as constitutive or differentially included ("annotated other splicing") and 654 (358) exons newly predicted in intronic regions ("novel exon"). Our analysis also showed that 29 of the 158 annotated MXEs are in fact not mutually exclusively spliced but represent constitutively spliced exons or other types of alternative splicing (Appendix Figs S2 and S3E). Finally, 1,741 (2,336) MXE candidates including 1,090 (1,402) newly predicted exons and 17 (29) of the annotated MXEs are supported by 0.5 million exon and 13 million SJ matching reads but still have to be regarded as MXE candidates because not all annotation criteria were fulfilled (Appendix Fig S3A and E).

To estimate the dependence of MXE confirmation and rejection on data quantity, we cross-validated the MXE gain (validation) and loss (rejection) events for several subsets of the total RNA-Seq data (Fig 1C, Appendix Fig S11, Materials and Methods "Saturation analysis"). The course of the curves provides strong evidence for the validity of the MXEs because a single exon-joining read would already be sufficient to reject an MXE cluster while at least two SJ reads are needed to validate one. Whereas even 15 billion RNA-Seq reads do not achieve saturation for the amount of validated MXEs, the gain in rejected MXE candidates is virtually saturated using 25% of the data.

To further validate the list of MXEs, we compared MXE clusters that contained two "annotated other splicing" exons to splicing information from GTEx portal (https://www.gtexportal.org/home/). Although GTEx portal uses an alternative aligner and different alignment settings, all MXEs that we compared showed mutually exclusive behaviour in GTEx portal (Appendix Fig S12), substantiating our results. Lastly, we selected six brain-expressed novel MXEs for qPCR validation in human brain total RNA. All assayed MXEs showed perfect coherence with the alignment results, confirming mutually exclusive splicing of all assayed novel MXEs in human brain (Appendix Fig S13, Dataset EV3).

Many of the 1,399 (855) MXEs have roles in the cardiac and muscle function and development, while cassette exons are enriched for microtubule- and organelle localization-related terms (Appendix Fig S14).

In summary, the high-confidence set of 1,399 (855) MXEs extends current knowledge of human MXE usage by an order of magnitude, (re)-annotating over a thousand existing and predicted exons and isoforms, while suggesting the existence of further human MXEs.

## The human genome contains large cluster and multi-cluster MXEs

In general, mutually exclusive splicing can be quite complex. This is best demonstrated by genes in arthropods that contain both multiple MXE clusters ("multi-cluster") and large clusters with up to 53 MXEs

such as in the *Drosophila Dscam* genes (Graveley *et al*, 2004; Pillmann *et al*, 2011). This is in strong contrast to mutually exclusive splicing in vertebrates as there is to date no evidence of multi-cluster or higher order MXE clusters (Matlin *et al*, 2005; Pan *et al*, 2008; Wang *et al*, 2008; Gerstein *et al*, 2014; Abascal *et al*, 2015a,b).

The analysis of the 1,399 validated human MXEs provides first evidence for clusters of multiple MXEs in the human genome (Fig 1D, Appendix Fig S15). While most MXEs are present in clusters of two exons (1,116 MXEs), a surprisingly high number of clusters have three to 10 MXEs (283 MXEs in 71 clusters).

Interestingly, although a large part of the verified MXEs contain a single MXE cluster (554 genes, Fig 1E), we could also provide evidence for human genes containing multiple MXE clusters. Thus, *TCF3*, *NEB*, *ANKRD36C* and *MTHFD1L* contain three clusters and *TTN*, *CAMK2D* and *CUX1* four clusters of MXEs. A very interesting case of complex interleaved mutually exclusive splicing can be seen for *CUX1*, the transcription factor cut-like homeobox 1. It contains a cluster of MXEs (exons 3b–3e) that is differentially included into a set of two exons (exon 3 and exon 4), and the two sets are themselves mutually exclusive (Fig 1F, Appendix Figs S16 and S17). The identification of large clusters with multiple MXEs and many genes with multiple clusters shows that complex mutually exclusive splicing is not restricted to arthropods (Schmucker *et al*, 2000; Graveley, 2005; Lee *et al*, 2010; Hatje & Kollmar, 2013) but might be present in all bilateria.

## Mutually exclusive presence of coding exons in functionally active transcripts

To understand which splicing mechanisms might be primarily responsible for the regulation of mutually exclusive splicing in humans, we investigated several mechanisms that were shown to act in some specific cases and were proposed to coordinate mutually exclusive splicing in general (Fig 2A; Letunic *et al*, 2002; Smith, 2005). We identified five cases (0.79% of all clusters) of U2 and U12 splice acceptor incompatibility (Appendix Fig S18) and 57 (9%) cases of potential steric interference, a too short distance between splice donor sites and branch points (< 50 bp; Fig 2B and Appendix Fig S19). Although 377 (60%) of the MXE clusters contain exons with exon lengths not divisible by three which would result in non-functional transcripts in case of combined inclusion, MXE-joining reads were found for only 83 (22%) of these clusters (Fig 2B; Appendix Figs S3B and D, and S20). Surprisingly, the majority of the annotated MXEs are of this type (91 of 122; 75%) as well as many exons previously annotated as other splice types (44 of 662), but only few of the novel MXEs predicted in intronic regions (25 of 615; Appendix Fig S3A and D). These numbers suggest that splicing of the remaining 484 MXE clusters is tightly regulated by other mechanisms (Fig 2B) such as RNA–protein interactions, interactions between small nuclear ribonucleoproteins and splicing factors (Lee & Rio, 2015), and competitive RNA secondary structural elements (Graveley, 2005; Yang *et al*, 2012; Lee & Rio, 2015). Competing RNA secondary structures are, however, usually not conserved across long evolutionary distances. A potential case of a docker site and selector sequences downstream of each exon variant was identified for the cluster of four MXEs in the *CD55* gene (Appendix Fig S21).

In contrast to cassette exons and micro-exons, which tend to be located in surface loops and intrinsically disordered regions instead

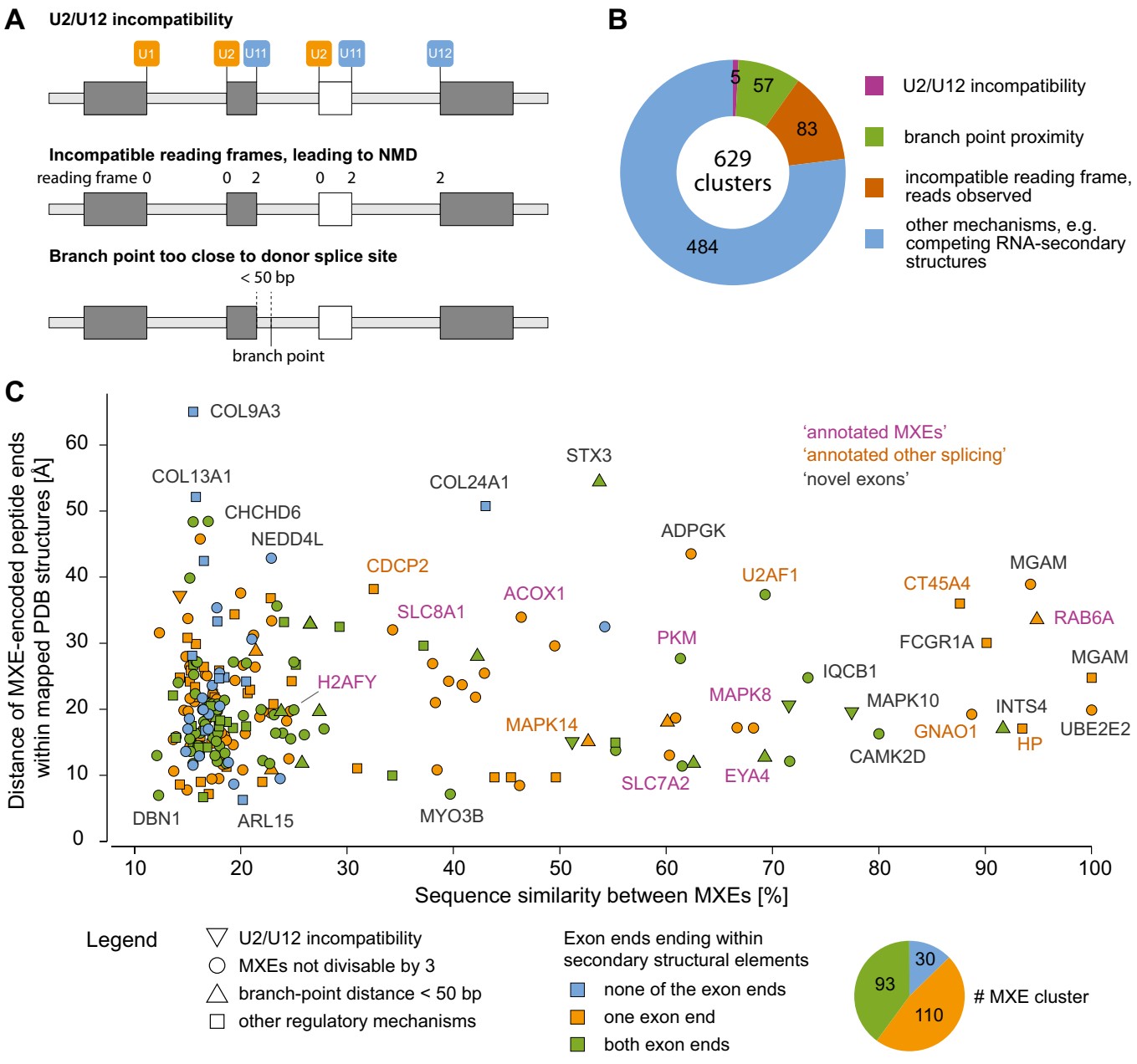

**Figure 2. MXE presence is regulated at the RNA and protein folding level.**

A   Schematic representation of MXE splicing regulation via splice-site incompatibility, branch point proximity and translational frame shift leading to NMD.

B   Observed usage of MXE splicing regulation in 629 MXE clusters.

C   By mutually exclusive inclusion into transcripts, MXEs of a cluster are supposed to encode the same region of a protein structure. If the respective regions of the protein structures are embedded within secondary structural elements (the ends of the exon-encoded peptides are part of α-helices and/or β-strands), it is highly unlikely that the translation of a transcript will result in a folded protein in case the respective exon is missing (skipped exon). If the MXEs have highly similar sequences and do not encode repeat regions, it seems unlikely that either could be present in tandem or absent at all in a folded protein. Here, we have combined protein structure features (colours) with splicing regulation information (symbols). Accordingly, 87% of the MXE-encoded protein regions are embedded in secondary structural elements (orange and green symbols), and most of the remaining MXEs can only be spliced mutually exclusive because splicing as differentially included exons would lead to frame shifts (blue circles). As examples, we labelled many MXE clusters distinguishing annotated MXEs (purple letters), known exons that we validated as MXEs (orange letters), and clusters containing novel exons (dark-grey letters).

of folded domains (Buljan *et al*, 2012; Ellis *et al*, 2012; Irimia *et al*, 2014), all MXEs, whose protein structures have been analysed, are embedded within folded structural domains as has been shown for, for example, *DSCAM* (Meijers *et al*, 2007), *H2AFY* (Kustatscher *et al*, 2005), the myosin motor domain (Kollmar & Hatje, 2014) and *SLC25A3* (Tress *et al*, 2017a). As we have shown in the beginning, there is also a subset of 73 MXEs not showing any sequence homology ("annotated no similarity"). It is unlikely that the encoded

peptides account for identical secondary structural elements. Rather, if the MXEs of this subset are true MXEs, there is a small subset (about 5%) of MXEs whose mutual inclusion leads to considerably altered protein folds or affects surface loops and disordered regions similar to cassette exons.

Because MXEs are supposed to modulate protein functions through variations and not alterations in specific restricted parts of the structure, we thought it could be possible to distinguish MXEs from cassette exons at a protein structural level. Such an analysis could provide complementary evidence for the validation as MXE in contrast to two (or more) neighbouring cassette exons. While one and only one of the exons of a cluster of MXEs has to be included in the transcript, the defining feature of a cassette exon is that it can either be present or absent. If MXEs were mis-classified and in fact neighbouring cassette exons, it would therefore be possible that all exons of the cluster were present or absent from the transcript, and accordingly the protein structure. These differences between MXEs and cassette exons impose three restrictions on their localization within protein folds (Appendix Fig S22). Thus, (i) if one or both ends of the MXE-encoded peptide end within a secondary structural element, it seems impossible that the respective peptide could be absent from the protein because this would break up multiple spatial interactions. This suggests that respective protein regions cannot be encoded by cassette exons. (ii) High sequence similarity between MXEs suggests important conserved structural interactions even if the peptide ends are not part of secondary structural elements. For example, it seems highly unlikely that a cluster of two exons encoding transmembrane helices could be spliced as cassette exons because absence or presence of both exons would switch the membrane site of all subsequent sequence. (iii) In case of cassette exons and absence of the exons, it must be possible that the remaining sequence still folds correctly. This can be assessed if a protein structure is available with the respective exon-encoded region present. Supposing the respective region was absent, the remaining ends would need to be joined to result in a correctly folded domain, which seems extremely unlikely if the peptide ends are far apart. Such regions are also more likely encoded by MXEs. To assess this model, we mapped the validated MXEs against the PDB database (Fig 2C, Appendix Fig S22, Dataset EV4; Rose *et al*, 2015). Of the 1,399 MXEs, 273 MXEs (20%) from 233 MXE clusters (37%) matched to human or mammalian protein structures (Appendix Fig S22). For 87% of these MXEs, at least one of the exon termini is embedded within a secondary structural element, suggesting that these exons are in fact true MXEs and not mis-classified cassette exons (Fig 2C, yellow and green coloured symbols). This high level of structural conservation also strongly supports the hypothesis that MXEs modulate but do not considerably alter protein functions (Letunic *et al*, 2002; Yura *et al*, 2006; Abascal *et al*, 2015a; Tress *et al*, 2017a). Of the remaining 13% (Fig 2C, blue coloured symbols), many MXEs would lead to frame shifts if they were spliced as cassette exons (both exons present or absent in the transcript, blue circles), and in multiple cases (e.g. *COL9A3*, *COL24A1* and *COL13A1*), the peptide ends are far apart indicating strong folding problems in case the respective exons were absent in the transcripts. In total, there are only a handful cases such as the MXE cluster in *ARL15* (Fig 2C) whose mutually exclusive presence in proteins cannot be explained by the analysed splicing restrictions, by NMD targeting, or by folding constraints.

## MXEs mainly consist of one ubiquitous exon and otherwise regulated exons

To modulate gene functionality, mutually exclusive splicing would need spatial and temporal splicing regulation and expression. To understand the expression patterns of MXEs, we conducted a differential inclusion analysis using the Human Protein Atlas (Fagerberg *et al*, 2014), Embryonic Development (Yan *et al*, 2013) and ENCODE datasets (Djebali *et al*, 2012). Of the 1,399 MXEs, 608 MXEs (345 unique genes), 573 MXEs (389 unique genes) and 552 MXEs (330 unique genes) are differentially expressed, respectively (adjusted *P*-value < 0.05; Fig 3A, Appendix Figs S23–S26, Dataset EV5 and EV6). Most notably, the differentially included MXEs comprise 43.5, 40.9 and 39.5% of all MXEs indicating that MXEs are to a very large extent tissue- and developmental stage-specifically expressed.

The comparison of the genes containing differentially expressed MXEs from these three projects shows that 519 (88.7%) of all 585 MXE cluster containing genes have at least a single MXE differentially expressed in one of the covered tissues, cell types or developmental stages (Fig 3B). The 519 genes contain 942 differentially expressed MXEs (67% of the total 1,399 MXEs; Fig 3C). This number is in agreement with earlier analyses on small sets of MXEs (66 and 57%) (Wang *et al*, 2008; Abascal *et al*, 2015a). Expectedly, the expression of novel MXEs seems to be considerably more tissue specific than the expression of annotated MXEs and cassette exons (Appendix Fig S23). Lastly, 208 MXEs from 113 genes are preferentially expressed during embryonic development indicating that many MXEs are specific to certain developmental stages (Fig 3B and C).

The analysis of MXE specificity reveals that in many clusters one MXE dominates expression, whereas other MXEs are expressed at selected developmental time points and in specific tissues (Fig 3, Appendix Figs S23–S26). This modulation suggests crucial spatio-temporal functional roles for MXEs and can in many cases not be observed at the gene level, as gene counts can remain largely invariant. A well-known case for similar expression of MXEs in newborn heart but expression of only one MXE variant in adult heart is the ion channel *CACNA1C* (Diebold *et al*, 1992), an example for the switch of expression are the MXEs of the *SLC25A3* gene (Wang *et al*, 2008). We surmise that the observed specificity in combination with a generally lower expression could also explain the discovery of 654 (358) novel exons that have so far eluded annotation efforts (Fig 1A, Appendix Fig S23). In conclusion, the tight developmental and tissue-specific regulation of MXE expression suggests that changes in MXE function or expression might cause aberrant development and human disease (Xiong *et al*, 2015). Pathogenic mutations in MXEs are known to cause Timothy syndrome, cardiomyopathy, cancer and kidney disease (Kaplan *et al*, 2000; Splawski *et al*, 2004, 2005; David *et al*, 2010; Mayr *et al*, 2011).

## MXEs are high-susceptibility loci for pathogenic mutations

To obtain a comprehensive overview of MXE-mediated diseases, we annotated all MXEs with pathogenic SNPs from ClinVar (Landrum *et al*, 2016), resulting in 35 MXEs (eight newly predicted exons) with 82 pathogenic SNPs (Fig 4A, Dataset EV7). Disease-associated MXEs show tight developmental and tissue-specific expression with

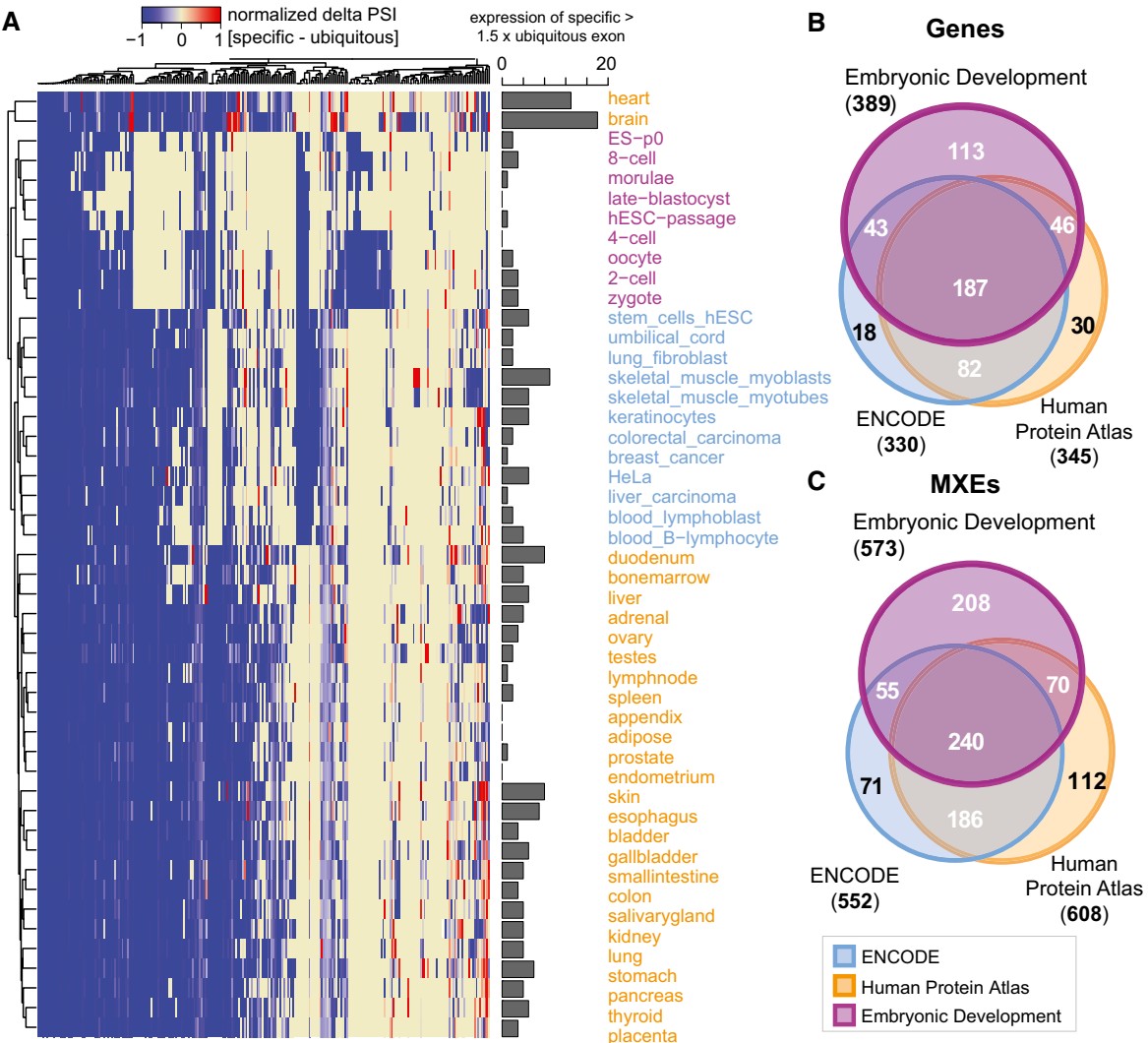

**Figure 3. MXE expression is tightly regulated across tissues and development.**

A   Heatmap showing all differentially expressed MXE clusters with at least three RPKM. Here, we used the Gini coefficient, which is a measure of the inequality among values of a frequency distribution (Ceriani & Verme, 2012) and has successfully been used to determine tissue-enriched gene sets (Zhang *et al*, 2017), to determine highly tissue-specific MXEs (maximum normalized Gini index of cluster) and MXEs with a broad tissue expression distribution (minimum Gini index). For each MXE cluster, the per cent-spliced-in (PSI) value of the ubiquitous MXE (minimum Gini index) is subtracted from the PSI value of the specific MXE (maximum Gini index of cluster) (delta PSI value) and scaled between −1 (broad tissue distribution) and 1 (highly tissue specific). Each column represents an MXE pair, and each row represents MXE expression in a tissue, cell type or at a developmental time point. The bar graph summarizes counts where the specific MXE is 1.5-fold more spliced in than the ubiquitous MXE.

B   Overview of differentially expressed genes for the Embryonic Development, ENCODE and Human Protein Atlas datasets.

C   Overview of differentially expressed MXEs for the Embryonic Development, ENCODE and Human Protein Atlas datasets.

prominent selective expression in heart and brain, and cancer cell lines (Fig 4B and C, Dataset EV7). Interestingly, the percentage of pathogenic SNP-carrying MXEs is twofold higher than the percentage of all pathogenic SNP-carrying exons (Fisher's exact test, *P*-value = $3 \times 10^{-11}$). A similar enrichment can be found for cassette exons (Fisher's exact test, *P*-value = $2.2 \times 10^{-16}$) suggesting that in general alternative splicing-associated exons are susceptibility loci for pathogenic mutations. The genes with MXEs carrying pathogenic SNPs are predominantly associated with neurological disease (10), neuromuscular disorders (7), cardiomyopathies (6) and cancer (3) and are enriched in voltage-gated cation channels

(e.g. *CACNA1C* and *CACNA1D*), muscle contractile fibre genes (e.g. *TPM1*), and transmembrane receptors (e.g. *FGFR1-3*; Fig 4, Appendix Fig S27, Dataset EV7).

Disease-associated MXEs have high amino-acid identity (average 49.1%, SD 23.1%), reaching up to 89% in *ACTN4* (Appendix Fig S28), suggesting similar functional roles and in consequence similar pathogenic potential for many MXE pairs (Fig 4C, Appendix Fig S29). Four of all SNP-containing MXE clusters contain mutations in both MXEs (*FHL1*, *MAPT*, *CACNA1C* and *CACNA1D*), whereas 31 currently have pathogenic SNPs in only one MXE. The MXE expression analysis shows that many SNP-carrying MXEs are highly

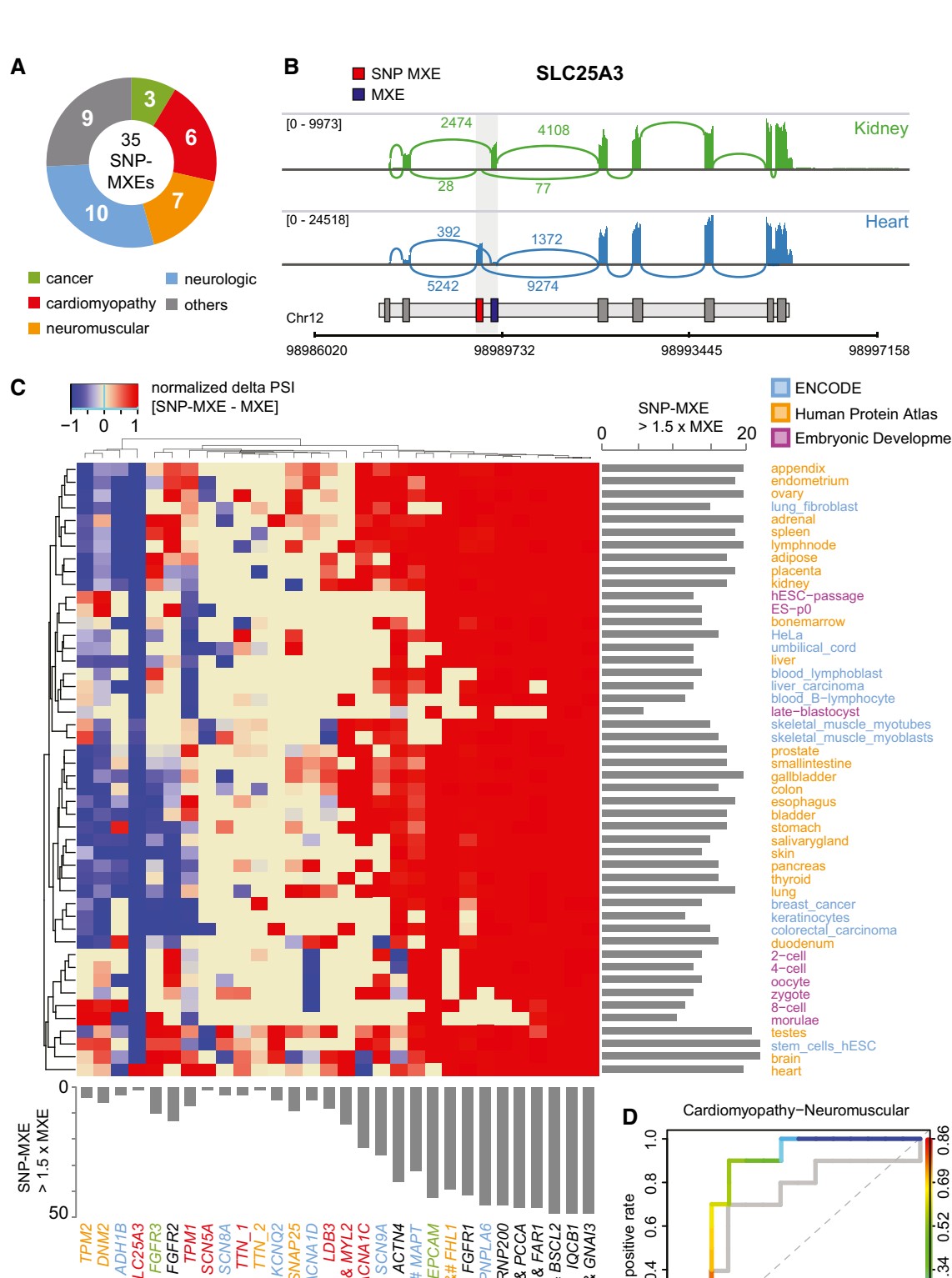

**Figure 4.**

**Figure 4. MXE-ratio expression predicts disease pathology.**

A   Thirty-five MXE clusters contain 82 pathogenic mutations causing neurologic (10), neuromuscular (7), cardiac (6), cancer (3) or other diseases (9).
B   Sashimi plots showing exon as well as splice junction reads (including number of reads) in kidney and heart for *SLC25A3*.
C   Heatmap showing the delta PSI values (PSI value of the non-SNP-containing MXE subtracted from the PSI value of the SNP-containing MXE) of MXE clusters containing pathogenic SNPs scaled between −1 and 1 (blue = high expression non-SNP-containing MXE, red = high expression SNP-containing MXE). Columns represent MXE clusters and rows tissues, cell types and developmental stages. The column bar graph summarizes counts where the SNP-containing MXE is 1.5-fold more expressed than the non-SNP-containing MXE, whereas the row bar graph shows this for each tissue, cell type and developmental stage.
D   Receiver-operating characteristic (ROC) curve showing true- and false-positive rates for cardiomyopathy-neuromuscular disease prediction based on spatio-temporal MXE (coloured lines and black text) and RPKM-based gene (grey lines and text) expression (delta PSI values).

expressed, especially in disease-associated tissues where the respective non-SNP-carrying MXEs are not or barely expressed (Fig 4B and C, Appendix Fig S29). Examples include *ACTN4*, *TPM1* and *SLC25A3* (Appendix Figs S28, S30, and S31). Moreover, MXEs with pathogenic SNPs are usually not or non-exclusively expressed at early developmental stages (Appendix Fig S28–S31), while high and exclusive expression could lead to early embryonic death or severe multi-organ phenotypes (e.g. *FAR1*, Appendix Fig S32). Conversely, several non-SNP-carrying MXEs are highly expressed in early development and are otherwise mainly expressed at equal and lower levels compared to the SNP-carrying MXEs (Appendix Figs S29E–S31). The absence of pathogenic SNPs in these MXEs suggests functional compensation of the pathogenic SNP-carrying MXEs or early lethality, both of which would result in no observable phenotype.

Of the 35 MXE clusters with pathogenic mutations eight contain novel exons (Fig 4C, Dataset EV7). A mutation in exon 9a (p.Asp365Gly) of *FAR1*, a gene of the plasmalogen–biosynthesis pathway, causes rhizomelic chondrodysplasia punctata (RCDP), a disease that is characterized by severe intellectual disability with cataracts, epilepsy and growth retardation (Buchert *et al*, 2014). Novel MXE 9b is expressed in the same tissues but at eightfold lower levels suggesting partial functional compensation of the MXE 9a mutation, which might be responsible for the "milder" form of RCDP as compared to pathogenic mutations in other genes of the pathway (*PEX7*, *GNPAT* and *AGPS*) (Appendix Fig S32). A tissue-specific compensation mechanism had already been proposed but a reasonable explanation could not be given because *FAR2* expression shows a different tissue profile and individuals with deficits in peroxisomal β-oxidation, a potential alternative supply for fatty alcohols, have normal plasmalogen levels (Buchert *et al*, 2014). Because of the young age of the affected children, it is not known yet whether a mutation in constitutive exon 4 (p.Glu165_Pro169delinsAsp), which could not be compensated in a similar way as the exon 9a mutation, leads to a strong RCDP-like phenotype (no survival of the first decade of life) or to a milder form such as the one caused by the exon 9a mutation.

In conclusion, it is tempting to speculate that MXE pathogenicity might be governed by high or exclusive expression in affected target tissues that is usually absent from early developmental processes, a pattern of expression that seems at least partially inversed for MXEs without pathogenic SNP annotations. To assess whether MXE pathogenicity follows observable rules, we trained a machine learner on MXE expression data and predicted the affected target tissue (Fig 4D, Dataset EV8). To obtain at least 10 observations per category with an expression > 3 RPKM, diseases were grouped into cardio-neuromuscular ($n = 10$) and other diseases ($n = 14$) and predicted using leave-one-out cross-validation with a Random Forest. Cardiac-neuromuscular diseases could be predicted with an accuracy of 83% (*P*-value < 0.01), a specificity of 79%, a sensitivity of 90% and an area under the ROC curve (AUC) of 85% (Fig 4D, Dataset EV8, Appendix Fig S29). Conversely, cardiac-neuromuscular disease could be predicted with an AUC of 72% using RPKM-based gene expression values (Fig 4D). Although based on only 24 observations, our data suggest that MXE expression might predict disease pathogenicity in space and potentially also in time.

## Evolutionary dynamics of MXEs in mammals and bilaterians

While tissue-specific gene expression is conserved between birds and mammals, the alternative splicing of cassette exons is conserved only in brain, heart and muscles and is mainly lineage-specific (Barbosa-Morais *et al*, 2012; Merkin *et al*, 2012). Accordingly, a core set of only ~500 exons was found with conserved alternative splicing in mammals and high sequence conservation, which was a small subset of the thousands of cassette exons identified in total. In contrast, although the total number was considerably smaller, most of the known human MXEs have been shown to be highly conserved throughout mammals if not even vertebrates (Letunic *et al*, 2002; Copley, 2004; Abascal *et al*, 2015b). In order to assess the conservation of human MXEs across mammals, we identified orthologous proteins in 18 representative species from all major sub-branches spanning 180 million years of evolution and predicted MXEs therein (Fig 5, Appendix Fig S33, Dataset EV9). Based on a

**Figure 5. Evolutionary dynamics of MXEs in mammalian evolution.**

Clusters of validated MXE were sorted by chromosome and chromosomal position. The names of the corresponding genes and the cluster-IDs are given in the outermost circle, and the presence of the respective MXEs (MXE clusters) in other annotations and mammals is indicated by coloured bars. Because the generation of the set of MXE candidates was based on the GenBank annotation, we analysed the presence of the validated MXEs in complementary annotations. Thus, the outer circles show whether the validated MXEs are also annotated as MXEs in Ensembl and Aceview, and whether the validated MXEs are present at all as exons in the Ensembl annotation as indicated by the legend. The lengths of the bars denote the percentage of matching exons for each cluster. For comparison, we show the annotation as MXE in two different Ensembl versions highlighting the dynamics of exon annotations over time. The comparison of the GenBank with the latest Ensembl annotation (v. 37.75) showed considerably less exons annotated as MXEs (58) in Ensembl although these include six of the "novel exons" (Appendix Fig S1). The presence of the respective validated MXEs in each of the analysed 18 mammals is shown by coloured bars. The 18 mammals, their phylogenetic relation and the total numbers of MXEs shared with human are presented at the bottom. The innermost circle represents the number of exons within each cluster of MXEs.

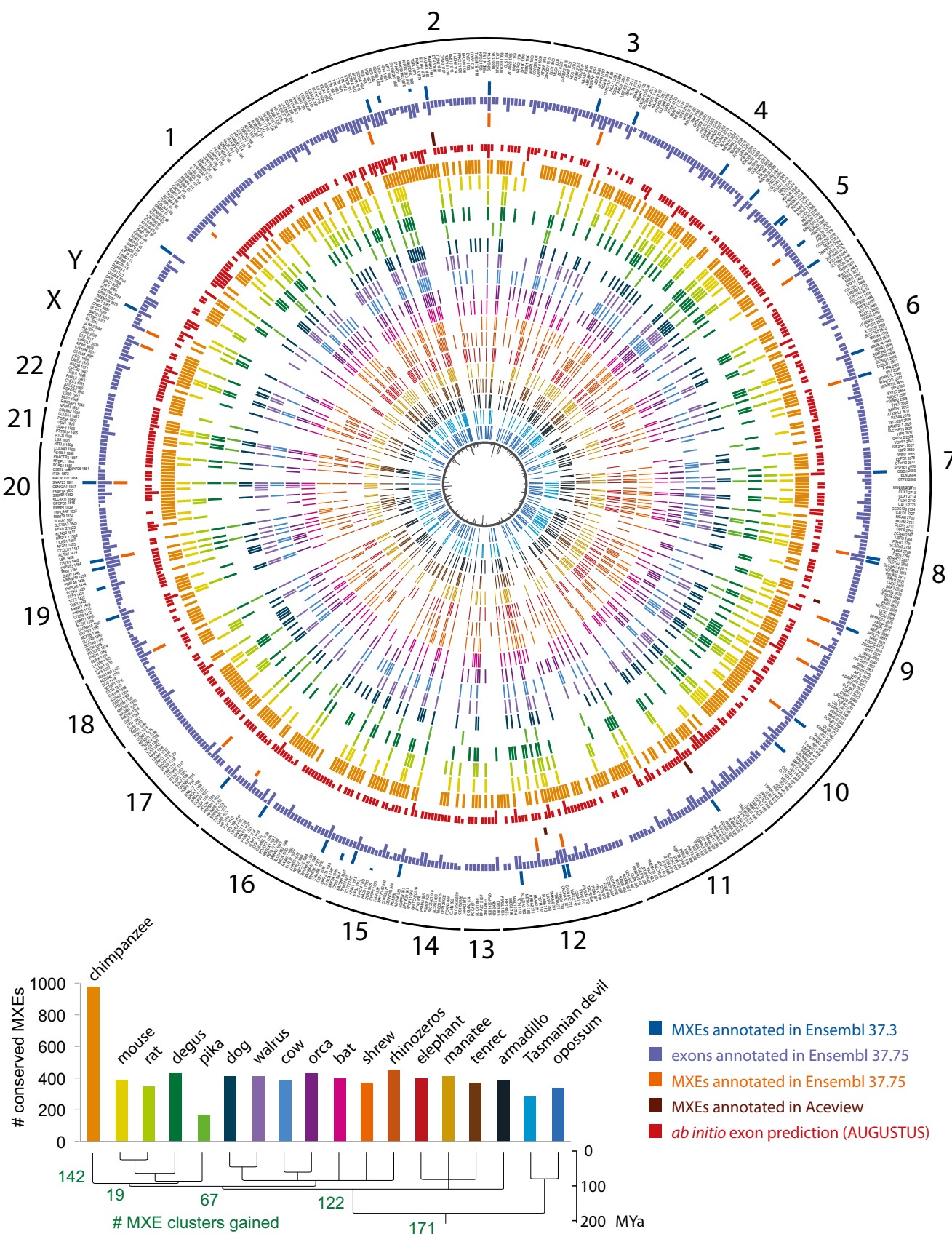

**Figure 5.**

simple model expecting each shared cluster to be already present in the last common ancestor of the respective species, we identified a core set of at least 173 (28%) of the human MXE clusters conserved throughout mammals (Fig 5, Appendix Fig S33). Other 122 MXE clusters were most likely present in the last common ancestor of the eutherians (16 species, placental mammals). The core set of mammalian MXE clusters includes 83 clusters shared between at least 16 of the species and 61 clusters shared between 17 species suggesting that their spurious absence in single mammals is likely due to genome assembly gaps or problems in identifying the correct orthologous genes. The remaining 29 MXE clusters of the core set have a scattered distribution across the 18 mammals indicating multiple independent branch- and species-specific cluster loss events. Such taxon-specific loss events include the MXE clusters in the *SRPK1* and *PQBP1* genes, which are absent in Glires (including mouse and pika), the cluster of 10 MXEs in *ABI3BP* that has been lost in the ancestor of mouse and rat, and the MXEs in *OSTF1* and *PTPRS*, which are absent in Afrotheria. The MXE clusters in *IKZF3*, *MBD1* and *ATP10B*, for example, are present in all Eutheria but not in Metatheria (marsupials). The MXE cluster gain rate within eutherian evolution towards human is relatively constant over time with about 23 clusters per 10 million years. Interestingly, each of the 16 eutherian species also lost a similar number of MXE clusters (127 clusters on average, Appendix Fig S33). In total, 82% of the human clusters containing validated MXEs are found in at least one further mammal (Fig 5). In summary, the large core set of mammalian MXEs and the overall conservation of MXE clusters suggest that MXEs are considerably more conserved than cassette exons. This observation supports expectations from considering the encoded protein structures where MXEs are supposed to provide alternative sequences for conserved secondary structural elements, while cassette exons are on average considerably shorter and add flexibility to surface loops (Buljan *et al*, 2012; Ellis *et al*, 2012; Irimia *et al*, 2014).

To get a first glimpse on mutually exclusive splicing evolution across bilaterians, we identified a set of 44 orthologous genes from genes containing MXEs in *Drosophila* (Hatje & Kollmar, 2013) and human genes containing MXE candidates (Appendix Fig S34, Dataset EV10). Of these orthologous genes, 28 contain validated MXEs in human, nine were validated to be spliced differently in human, and seven could not be validated in human because read mapping data are still missing; 20 (71%) of the genes containing validated MXEs represent cases of incompatible reading frames leading to NMD in case of joined inclusion, and for 18 of these MXE clusters multiple MXE-joining reads were found (Appendix Figs S34 and S35). We further analysed the 28 orthologous genes with validated MXEs and found five genes with homologous MXE clusters (identical position in gene, identical exon phase), 13 genes with MXE clusters in human that have homologous exons in *Drosophila* and eight genes with MXEs in human where the corresponding sequence regions in the orthologous *Drosophila* genes are part of larger exons (Appendix Figs S35 and S36). The presence of orthologous MXE clusters has been attributed to convergent evolution (Copley, 2004), although the respective analysis was in part based on the comparison of non-orthologous genes (e.g. comparing human sodium channel genes [e.g. *SCN1A*] with the *Drosophila* calcium channel *cac* gene and not the orthologous sodium channel *para* gene). At least for muscle myosin heavy chain genes it could

be demonstrated that *Drosophila* already lost several MXE clusters compared to, for example, *Daphnia pulex* (crustacean) and lophotrochozoans (Kollmar & Hatje, 2014) and that the evolutionary history of the MXEs within each cluster is remarkably complex with multiple independent exon duplications and losses (Odronitz & Kollmar, 2008). Thus, detailed studies including more bilaterian and non-bilaterian taxa would be necessary to finally conclude convergent or divergent evolution for each of the human and *Drosophila* MXE clusters. Although the overlap of MXEs in orthologous genes of human and *Drosophila* is very low, the MXE gain and loss rates are very similar (Hatje & Kollmar, 2013) indicating a conserved role of tandem exon duplication in bilaterians. Gene structures can be highly conserved between kingdoms (Rogozin *et al*, 2003), and certain exons therefore seem to be predisposed to undergo duplication. In summary, these findings provide strong evidence for many MXE gain and loss events during mammalian evolution, suggesting a pronounced role of these processes in speciation and establishing phenotypic differences.

## Discussion

Using stringent criteria, including sequence similarity, reading frame conservation and similar lengths, and billions of RNA-Seq reads, we generated a strongly validated atlas of 1,399 human MXEs providing insights into mutually exclusive splicing mechanics, specific expression patterns, susceptibility for pathogenic mutations and deep evolutionary conservation across 18 mammals. The presented increase in human MXEs by an order of magnitude lifts MXEs into the present-day dimension of other human alternative splice types (Pan *et al*, 2008; Wang *et al*, 2008; Gerstein *et al*, 2014). Saturation analysis and the existence of 1,816 expressed but unconfirmed MXE candidates suggest a potential 27% increase in the MXE-ome with a twofold increase in data. Although alternative splice variants are abundant at the transcriptome level, recent mass spectrometry analyses suggested only small numbers of alternative transcripts to be translated (Abascal *et al*, 2015a; Ezkurdia *et al*, 2015; Blencowe, 2017; Tress *et al*, 2017a,b). Interestingly, MXEs were particularly enriched in the translated alternative transcripts, compared to other splice variants. However, ribosome profiling data showed high frequencies of ribosome engagement of cassette exons indicating that these isoforms are likely translated (Weatheritt *et al*, 2016). Similar results have been obtained through polyribosome profiling (Sterne-Weiler *et al*, 2013; Floor & Doudna, 2016). These observations suggest that most of the MXEs evaluated at the transcript level will also be found in the proteome.

About half (47%) of the 1,399 MXEs represent novel exons, which are often expressed at low levels and whose expression is restricted to few tissues and cell types, possibly explaining their absence from current genome annotations. Extrapolating these observations to all splice types and genes suggests the existence of thousands yet unannotated exons in introns. This estimation is in accordance with a recent analysis of more than 20,000 human RNA-Seq datasets that revealed over 55,000 junctions not present in annotations (Nellore *et al*, 2016). In this analysis, junctions found in at least 20 reads across all samples were termed "confidently called". Although the total number of reads required for MXE validation in our analysis is lower (≥ 2 SJ reads in the 1SJ case, ≥ 6 SJ

reads in the 3SJ case), the numbers seem more conservative given that we used 40 times less data for the validation.

The almost 10-fold increase in the human MXE-ome supports recent suggestions that mutually exclusive splicing might play a much more frequent role than anticipated (Pan *et al*, 2008; Wang *et al*, 2008; Ezkurdia *et al*, 2012; Abascal *et al*, 2015a). By comparing differentially expressed MXEs across cell types, tissue types and development, we could show that 14% of all genes with MXE clusters are shared between the three data sources, and 39% between any two. Most notably, however, it is almost always a different MXE from the same cluster that is differentially expressed, and only 3.3% of the MXEs are differentially expressed in all three data sources. We believe that this indicates a high spatio-temporal regulation of all MXEs in two-exon and multi-exon clusters. We rarely observed switch-like expression with only one of the MXEs of each cluster present in each cell- or tissue type or developmental stage. Rather, one of the MXEs ("default MXE") of each cluster was present in most or all samples and the other MXEs were expressed in several selected tissues and developmental stages ("regulated MXEs") in addition to the default MXE. Although the "regulated MXE" is usually expressed at lower level compared to the "default MXE", there is almost always at least a single tissue or developmental stage where it is expressed at higher level. This supports previous assertions on the modulatory and compensatory effects of the regulated MXE on the enzymatic, structural or protein interaction functions of the affected protein domains (Letunic *et al*, 2002; Tress *et al*, 2017a).

The concerted annotation and splicing analysis of novel exons have deep implications for the detection and interpretation of human disease (Bamshad *et al*, 2011; Gonzaga-Jauregui *et al*, 2012; Xiong *et al*, 2015; Bowdin *et al*, 2016). For one, exome and panel sequencing remains the method of choice for the detection of genetic diseases and both methods rely on current exon annotations (Chong *et al*, 2015). Furthermore, our data suggest that MXE expression might reflect disease pathogenesis that could allow for the prediction of the affected organ(s). It is intriguing to speculate that the observed expression–disease association is a general dogma, which could be used to predict yet unseen diseases from published expression data, potentially bringing about a paradigmatic shift in (computational) disease research.

# Materials and Methods

## Data sources

The human genome assembly and annotated proteins (all isoforms) were obtained from GenBank (v. 37.3) (Benson *et al*, 2013). For MXE candidate validation, we selected data from 515 publically available samples comprising 31 tissues and organs, 12 cell lines and seven developmental stages (Barbosa-Morais *et al*, 2012; Djebali *et al*, 2012; Tilgner *et al*, 2012; Xue *et al*, 2013; Yan *et al*, 2013; Fagerberg *et al*, 2014) amounting to over 15 billion RNA-Seq reads. The data were chosen to encompass common and rare potential splice events in a broad range of tissues, cell types and embryonic stages. These RNA-Seq data were obtained from either GEO (NCBI) or ENA (EBI) databases (Dataset EV1). The description of the respective tissues and developmental stages is also listed in Dataset EV1.

## Reconstruction of gene structures

The gene structures for the annotated proteins were reconstructed with Scipio (Keller *et al*, 2008; Hatje *et al*, 2013) using standard parameters except `–max_mismatch=7`, `–region_size=20000`, `–single_target_hits`, `–max_move_exon=10`, `–gap_ to_close=0`, `–blat_oneoff=false`, `–blat_score=15`, `–blat_identity=54`, `–exhaust_align_size=20000`, and `–exhaust_gap_size=50`. We let Scipio start with `blat_ tilesize=7` and, if the entire gene structure could not be reconstructed, reduced the `blat_tilesize` step by step to 4. All parameters are less stringent than default parameters to increase the chance to reconstruct all genes automatically.

## Predicting mutually exclusive spliced exons

The human genome annotation does not contain specific attributes for alternative splice variants and thus does not allow extracting or obtaining lists for specific splice types. As mutually exclusive spliced exons (MXEs), we regarded those neighbouring exons of a gene locus that are present in only one of the annotated splice variants. These MXEs were termed "annotated MXEs". However, exons appearing mutually exclusive are not necessarily spliced as MXEs. Terminal exons, for example, are included in transcripts by alternative promoter usage and by alternative cleavage and polyadenylation. MXEs were predicted in the reconstructed genes using the algorithm implemented in WebScipio (Pillmann *et al*, 2011). The minimal exon length was set to 10 aa (`–min_exon_ length=10`). WebScipio determines the length of each exon ("search exon") and generates a list of potential exonic regions with identical lengths (to preserve the reading frame) within the neighbouring up- and downstream introns. To account for potential insertions, we allowed length differences between search exon length and potential new exonic region of up to 60 nucleotides in steps of three nucleotides [`–length_difference=20` (given in aa)], thus obtaining a list of "exon candidates". WebScipio then translates all exon candidates in the same reading frame as the search exon and removes all sequences that contain an in-frame stop codon. In case of overlapping exonic candidate regions, we modified the original WebScipio algorithm to favour exonic regions with GT–AG splice junctions over other possible splice sites (GC–AG and GG–AG). The translations of the exon candidates are then compared to the translations of the search exons, and candidates with an amino-acid similarity score of more than 10 (`–min_ score=10`) are included in the final list of MXE candidates. Because the exon candidate scoring is done at the amino acid level, WebScipio expects candidates for 5′ exons of genes to start with a methionine, and candidates for 3′ exons of genes to end with a stop codon. This minor limitation is due to WebScipio's original development as gene reconstruction software. MXE candidates for terminal exons were only searched in direction to the next/previous internal exon. The reason for looking for MXE candidates of annotated terminal exons is that we cannot exclude that further up- and downstream exons are missing in the annotation, which would turn the new MXE candidates to internal exons. Because of the described minor limitation, however, we can only propose MXE candidates if supposed additional up- and downstream exons are non-coding exons. Because terminal exons are

included in transcripts by alternative promoter usage and by alternative cleavage and polyadenylation, we treated the list of terminal exon candidates separately (Appendix Fig S4). This list might be of interest for further investigation for other researchers. Except for this Appendix Fig S4, we entirely focused on internal MXE candidates.

## Definition of criteria for RNA-Seq evaluation of the MXE candidates

While the sole mapping of RNA-Seq reads reveals the transcription of the respective genomic region, it does not prove the inclusion into functional transcripts. The mutually exclusive inclusion of the MXE candidates into functional transcripts requires at least the following splice junction (SJ) reads (Appendix Fig S5): (i) There must be SJ reads matching from every MXE to up- or downstream gene regions bridging the other MXEs of the cluster. The latter criterion takes into account that the annotated exons neighbouring the clusters of MXEs might not themselves be constitutive but alternative exons as, for example, in *NCX1* (Appendix Fig S6). (ii) SJ reads mapping from one to another MXE candidate lead to MXE candidate rejection except for those MXEs leading to a frame shift. Without this constraint, which has not been set in earlier analyses (Wang *et al*, 2008), MXEs cannot be distinguished from neighbouring differentially included exons, which are quite common in human (data not shown; see e.g. Hammesfahr & Kollmar, 2012 and Appendix Fig S6). Thus, there are three constraints for a cluster of two MXEs while clusters of three and five MXEs, for example, already require seven and 18 constraints, respectively (Appendix Figs S5 and S7). Under more stringent conditions, also SJ reads from MXEs to the neighbouring annotated exons independent of their splice type would be required giving rise to five constraints for a cluster of two MXEs (Appendix Fig S5).

Note that as a matter of principle the read coverage of MXEs and other alternative splicing events is considerably lower than that of constitutive exons due to their mutually exclusive inclusion in the transcripts. For example, each of the exons of a cluster of three MXEs is expected to only have, on average, one-third the coverage of the constitutive exons of the same gene. The number of predicted exons, of which both sites are supported by splice junction reads, is also considerably lower than the total number of supported MXE candidates (Appendix Fig S3), which we think is due to the general low coverage of the exons and not due to read mapping and exon border prediction problems (Appendix Fig S3).

## Validation of the MXE candidates by RNA-Seq mapping

SRA files were converted to FASTQ files using fastq-dump software (v. 2.1.18). FASTQ files were mapped onto the human reference genome (hg19) using the STAR aligner (v_2.3.0e_r291) (Dobin *et al*, 2013). To this end, we first generated a reference genome index with `–sjdbGTFfeatureExon`, `–sjdbGTFtagExonParentTranscript`, a splice junction overhang size of 99 (`–sjdbOverhang`) and GTF annotation files containing all transcripts and all MXE candidates. The MXE candidate GTF file was extracted from Kassiopeia database and is available for download there (Hatje & Kollmar, 2014). The mapping was done for each sample separately. We allowed a rather stringent maximum

mismatch of 2 (`–outFilterMismatchNmax 2`; STAR default is 10) and the output was forced to SAM format (`–outStd SAM`). Otherwise, default settings were used. The resulting files with the mapped reads were sorted, converted to BAM format and indexed with SAMtools (`sort -n`) for further processing (Li *et al*, 2009).

## Distinguishing MXEs from other splice variants

For the analysis of the read mapping data, we disassembled clusters with more than two MXE candidates into all possible sub-clusters. For example, a cluster with four MXE candidates [1,2,3,4] was fractionated into the following sub-cluster: [1,2], [2,3], [3,4], [1,2,3], [2,3,4], [1,2,3,4]. Each of these sub-clusters was analysed independently according to the validation criteria (splice junction reads present, exon-joining reads absent). If all criteria were satisfied for one of the sub-clusters, all MXE candidates of the respective sub-cluster were labelled "verified". In a second analysis, each cluster of MXE candidates was analysed for exon-joining reads, which denote constitutive splicing or splicing as differentially included exons. However, MXE candidates of clusters and sub-clusters with exon-joining reads but exon lengths not divisible by three were also flagged as "verified" because their combined inclusion would lead to a frame shift in the translation of the transcript.

## Limits of the MXE dataset

Similar to every genome annotation dataset, also the current dataset of RNA-Seq validated MXEs has some limitations. Some are inherent to the still incomplete human genome annotation that was used as basis for generating the list of MXE candidates. As mentioned above and shown in Appendix Fig S2C, there are genes with mis-annotated terminal exons overlapping MXEs. Also, there are "transcripts" in the GenBank dataset that combine exons from (now) different genes. The presence of these "transcripts" in the genome annotation might be the result of mis-interpreting cDNA data as coding sequence although these might be the result of some level of mis-splicing.

Similarly, mis-splicing might be an important reason for validating true MXEs as "non-MXEs". A single exon-joining read turns MXE candidates into non-MXEs, whose mutually exclusive splicing might otherwise be supported by thousands of MXE-bridging SJ reads. Given these limitations, we expect that many of the exons, that we currently tag as constitutive or other alternative splicing, might in fact be MXEs. On the other hand, our MXE dataset might also contain some exons that are in fact non-MXEs. This is well demonstrated in the saturation analysis (Fig 1C) showing that although more data will lead to the validation of many more exons as MXEs, for which SJ reads are currently missing, there will be clusters that will be rejected as soon as more data include exon-joining reads. In addition, some MXEs with only a few supporting SJ reads might in fact be pseudoexons. However, we also did not observe any SJ reads for about 15% of the annotated exons, which are nevertheless not regarded as pseudoexons (Fig 1B, Appendix Fig S3). Finally, some MXEs determined from transcripts showing complex splicing might in fact be mutually exclusive in transcripts, but not in the sense of a cluster of uninterrupted neighbouring exons.

## Saturation analysis

Theoretically, increasing the number of samples should also increase the number of validated MXEs, as the total increase in read number for different observed or novel tissues should increase the read evidence for the predicted MXEs. At the same time, increasing the number of reads also heighten the chance of rejecting an MXE candidate. This raises the question of what the expected number of validated and rejected MXEs for increasing numbers of samples is. Additionally, it would be interesting to obtain the theoretical point of saturation, the maximum expected number of MXEs in the human genome.

To obtain this information, sub-samples of STAR-aligned RNA-Seq splice junction (SJ) reads were used to estimate the expected recall and false-positive rate (Fig 1C, Appendix Fig S11). The number of verified MXEs was calculated using SJ reads for different percentages of the data. Similarly, the number of rejected MXEs was obtained. To reduce the bias from data sampling, datasets were chosen randomly and the saturation analysis was performed in 30 independent runs. To calculate the mean of validated and rejected MXEs at respective percentages of the total RNA-Seq data used for validation, we used the respective numbers from the 30 independent runs.

To estimate the potential increase in MXEs given more sequencing data, we fit the sub-sampling data to the number of expected MXEs $f(x)$ using Matlab and the optimal fits were obtained for a power function

$$f(x) = a * x^b + c$$

with the linear coefficient $a$, the exponential coefficient $b$ and the error term $c$ (Appendix Fig S11B). Given a twofold increase in the number of reads, the expected number of validated MXEs (1SJ) is $1{,}769 \pm 47$ (95% confidence interval), validated MXEs (3SJ) is $1{,}081 \pm 12$, rejected MXEs (1SJ) is $227 \pm 9$, and the number of rejected MXEs (3SJ) is $95 \pm 5$ (Appendix Fig S11B). While the number of validated MXEs is far from saturation (a 100% increase in data results in 27% increase in the number of validations), the number of rejected MXEs seems to be saturated (a 100% increase in data results in 2% increase in the number of rejections).

## qPCR validation of MXE candidates

Total RNA was purified from healthy human brain tissue (substantia nigra) using Trizol kit (Tri Reagent, Sigma T9424) following manufacturer's instructions. RNA was further purified using the RNA Clean & Concentrator © TM -5 kit (Zymo Research, cat. R1013). The RNA quality was investigated using the 6000 nano assay on a Bioanalyzer 2100 (Agilent Technologies). Reverse transcription was carried out using the iScript © cDNA Synthesis kit (cat# 1708890, Bio-Rad) using approximately 500 ng of total RNA in a volume of 20 μl.

Relative expression levels of the genes of interest as well as one housekeeping gene (glyceraldehyde 3-phosphate dehydrogenase [*Gapdh*]) were determined by qPCR using a LightCycler® 480. All qPCR experiments were performed in duplicates using SYBR™ Green PCR Master Mix (cat # 4309155). For each PCR, 20 ng cDNA

was used and negative controls contained no cDNA. The qPCR was run under the following conditions: pre-incubation at 95°C for 5 min, denaturation at 95°C for 10 s, annealing 60°C for 15 s, extension at 72°C for 10 s repeated for 40 cycles (Sybr green standard protocol II). Detailed information on the primers and qPCR results can be found in Dataset EV3.

## Analysis of the splice mechanism

To determine the distance between intron donor site and branch point, we analysed all introns smaller than 500 bp using the standalone version of SVM-BPfinder (beta) (Corvelo *et al*, 2010) to predict branch point locations. Longer introns harbour high numbers of branch point candidates, and the accuracy of the branch point prediction considerably decreases. Longer introns also often contain multiple branch points with different splicing kinetics (Corvelo *et al*, 2010) so that a steric hindrance criterion for splicing multiple MXEs into the same transcript might not apply anymore. Branch points are usually located in the 3′ regions of the introns and it seems highly unlikely to identify only a single potential branch point within an, for example, > 2,000-bp intron, which would in addition be located within the 5′ 50 bps. Thus, the highest-scoring location within the < 500-bp introns was taken as best guess for the branch point and the distance to the intron donor site determined.

In order to identify U12-type introns, we analysed all donor splice sites of the introns preceding the clusters of MXEs and those subsequent to all MXEs using the consensus pattern described by Sharp and Burge (Sharp & Burge, 1997). The acceptor splice sites of U12-type introns do not show conserved patterns and were therefore not used here for verification.

Binding windows for competing intron RNA secondary structures were predicted for all candidate clusters of MXEs using the SeqAn package (Döring *et al*, 2008). The identified binding windows of all homologous genes were aligned using MUSCLE (Edgar, 2004) and the RNA secondary structures predicted by RNAalifold (ViennaRNA package) (Lorenz *et al*, 2011).

## Mapping MXE sequences onto protein structures

To identify the best structural models for the sequences encoded by the MXEs, we mapped the protein sequences of the respective genes against available protein structure data. To this end, we made use of a recently developed database, called Allora (http://allora.motorprotein.de), in which genomic information is mapped onto protein structures. Allora currently contains 94,148 PDB entries (derived from the RCSB Protein Data Bank, http://www.rcsb.org, Rose *et al*, 2015) with 247,959 chains, of which 120,665 represent unique sequences. Based on the database references in the PDB entries, the full-length proteins were fetched from UniProt KB (UniProt Consortium, 2015) or GenBank (Benson *et al*, 2013) and the corresponding gene structures of the eukaryotic proteins reconstructed with WebScipio (Hatje *et al*, 2013). In Allora, all PDBs belonging to the same UniProt or GenBank entries are connected. BLAST+ (Camacho *et al*, 2009) was used to search for the most similar UniProt/GenBank protein sequence compared to the human proteins containing MXEs. The hit with the lowest E-value was taken, and the associated PDB chains were aligned to the human protein using m-coffee (Wallace *et al*, 2006). The MXE part of the alignment was

extracted for further analysis (=> "MXE structure"). As "intron distances", we determined the distances between the CA atoms of the first and the last residues of the MXE structures.

## Evaluating the differential inclusion of MXEs into transcripts

Splice junction read counts were extracted from STAR output "SJ.out.tab" files. For each MXE in a cluster, the per cent-spliced-in (PSI) value was calculated by dividing the number of junction reads of the MXE by the sum of junction reads for all MXEs in the same cluster. Differential inclusion analysis on the Human Protein Atlas, Embryonic Development and ENCODE datasets was performed using a Kruskal–Wallis rank sum test with a Benjamini–Hochberg (BH) multiple testing correction. Values were computed using the "kruskal.test" and "p.adjust" functions in R. For each project, we created a design matrix with sample name and experimental condition and replicate numbers. The results of the differential inclusion analysis are summarized in Dataset EV5.

## Differential expression of pairs of annotated and novel MXEs

For each sample (tissue, cell type and developmental stage), we calculated the median RPKM (reads per kilobase of transcript per million mapped reads) from the replicates for each MXE. To compile a set of MXEs with significant expression, only pairs of MXEs were selected of which either the annotated or the novel exon had a median expression of more than 3. The number of MXEs for this analysis would not considerably decrease if a cut-off of 30 were chosen (252 MXEs at a cut-off of 3 versus 240 MXEs at a cut-off of 30). For each pair of MXEs, we subtracted the PSI value of the ubiquitous/known/non-SNP-containing MXE from the PSI value of the respective specific/novel/SNP-containing MXE (delta PSI values) and scaled those values between −1 (high PSI for ubiquitous/known/non-SNP-containing MXE) and 1 (high PSI for specific/novel/SNP-containing MXE) (see also Figs 3A and 4C, Appendix Fig S23). In case an MXE pair was not expressed in a certain tissues (NA or 0), the value was set to 0.

## Inequality analysis

The mean PSI values of each MXE were calculated for each tissue in the Human Protein Atlas project, each developmental stage in the embryonic development (Peking University) project, and each cell type in the ENCODE (Caltech) project. For each MXE, the Gini index (Ceriani & Verme, 2012) was calculated independently for each project based on the mean PSI values using the Gini function with standard parameters from the ineq R package version 0.2-13 (Achim Zeileis, Christian Kleiber, https://CRAN.R-project.org/package=ineq; Cowell, 2011). For the analysis of MXE clusters, only those clusters were taken into account that include at least two MXEs with an RPKM ≥ 10 in at least one dataset within each project. Furthermore, we excluded clusters where all MXEs have "NA" PSI values within each project (244, 96 and 225 clusters, respectively).

## Identification of pathogenic SNPs in MXEs

To identify potentially pathogenic SNPs in MXEs, the MXEs were compared to the ClinVar SNP database (ClinVar VCF file

downloaded on 11 Aug 2016, version updated at 30 Jun 2016, Landrum *et al*, 2016). The ClinVar variant summary file (VCF file) was converted into a BED file keeping all original information. Positions overlapping between MXEs and ClinVar-SNPs were accessed using the BEDTools feature intersection software (Quinlan & Hall, 2010). SNPs are classified as pathogenic or non-pathogenic according to ClinVar's "ClinicalSignificance" field annotation. All entries containing "benign" and all structural variations were removed. All ClinVar-SNPs overlapping with MXEs were manually verified in order to keep only potentially pathogenic variations.

To access the statistical significance of disease enrichment in MXEs and cassette exons, we compared the amount of pathogenic SNP-containing to non-SNP-containing exons. Of 615,410 annotated exons, 21,030 (3.4%) contain pathogenic SNPs; of 1,399 MXEs, 99 (7.1%) contain pathogenic SNPs; and of 31,745 cassette exons, 2,143 (6.8%) contain pathogenic SNPs. The ~2-fold enrichment of alternative splicing-associated exons (MXEs and cassette exons) is highly significant (Fisher's exact test, $P$-value MXE $= 3 \times 10^{-11}$, $P$-value cassette $= 2.2 \times 10^{-16}$).

## Disease prediction using pathogenic SNPs in MXEs

In order to predict disease from MXE expression, we first filtered for MXEs that had a minimal RPKM value of 3 and then subtracted the expression of the non-SNP-containing MXE from the SNP-containing MXE for all MXE pairs with mutations, across all developmental stages, tissues and cell types (49 features per MXE pair). Delta PSI values (PSI for SNP-containing MXE—PSI for non-SNP-containing MXE) were subsequently scaled and centred, and the MXE pairs were annotated to two disease classes, cardiomyopathy-neuromuscular disease ($n = 10$) or other diseases ($n = 14$). We regrouped genes into these categories to obtain relatively balanced categories while keeping a minimum of 10 observations per category.

Classification with limited observations needs careful execution, as over-fitting (high variance) and under-fitting (high bias) are common problems. To avoid high variance or bias, several crucial steps were taken. First, we did not optimize hyperparameters, using a Random Forest with 250 trees and a maximum tree depth of 16 (number of predictors/3). Second, we used leave-one-out cross-validation to avoid sampling bias and model instability. Third, diseases were grouped into two categories of relatively even size (see above). Models were built using the R packages caret (Kuhn, 2008) and randomForest, and ROC curves were generated with ROCR (Sing *et al*, 2005).

Of note, models trained on PSI values (considering only the PSI value of the SNP-containing MXE, data not shown) or RPKM values (Appendix Fig S29) obtained similar accuracies as the model trained on delta PSI values, indicating the stability of the prediction across slight variations in feature pre-processing.

## Gene ontology enrichment analysis

We used WebGestalt for Gene Ontology enrichment analyses (Wang *et al*, 2013). The lists of unique genes in gene symbol format were uploaded to WebGestalt and the GO Enrichment Analysis selected. The entire human genome annotation was set as background and 0.05 as threshold for the $P$-value for the significance test using the

default statistical method "hypergeometric". Categorical enrichment of MXEs and cassette exons was summarized in a heatmap.

### Protein–protein interaction analysis

The protein–protein interaction network was built by using Gene-MANIA webservice (Warde-Farley *et al*, 2010). The list of unique genes containing a pathogen SNP was submitted to GeneMANIA's webservice, and we downloaded the resulting network in SVG format and manually included disease and ontology information.

### Assessing the dynamics of MXE annotations over time

MXEs might have already been annotated/described although not been included in the NCBI reference dataset. This might especially account for newer annotations based on the recently published ENCODE project data. Therefore, we obtained alternative protein sequence datasets from Aceview (Thierry-Mieg & Thierry-Mieg, 2006) and Ensembl (Yates *et al*, 2016). Further datasets like the VEGA and GENCODE annotations are continuously integrated into Ensembl and were therefore not considered separately. The Aceview database has been built in the year 2000 to represent comprehensive and non-redundant sequences of all public mRNA sequences. The human dataset has last been updated in November 2011, thus before the availability of the ENCODE data.

To assess the novelty of our MXE assignments with respect to the timely updates and changes of the human annotations, we compared our data with that of Aceview and with the latest annotation from Ensembl (Fig 5, Appendix Fig S1). As at the beginning of the project, only a few MXEs are annotated as such in other databases. Surprisingly, however, many of the previously annotated exons (independent of their splicing status) were removed from the latest Ensembl annotation, although our RNA-Seq mapping not only strongly supports their inclusion into transcripts but also their splicing as MXEs. This shows that further collaborative efforts are needed to reveal a stable and persistent human gene annotation.

### *Ab initio* exon prediction

Exon prediction by *ab initio* gene finding software is another means of generating a database of potential coding sequences. *Ab initio* exon prediction was done with AUGUSTUS (Stanke & Waack, 2003) using default parameters to find alternative splice forms and the feature set for *Homo sapiens*.

### Identifying orthologous proteins in 18 mammals

Cross-species searches in 18 mammals (Dataset EV9) were done with WebScipio (Hatje *et al*, 2013) with same parameters as for gene reconstructions except `–min_identity=60`, `–max_mismatch=0` (allowing any number of mismatches), `–gap_to_close=10`, `–min_intron_length=35`, `–blat_tilesize=6` and `–blat_oneoff=true`. MXE candidates in cross-species gene reconstructions were searched with `–length_difference=20`, `–min_score=15` and `–min_exon_length=15`, for all exons in all introns but not in up- and downstream regions. Reasons for not detecting clusters of MXEs might be gene and MXE loss events, sequence divergence precluding ortholog identification, and

assembly gaps. For determining the origin of a conserved MXE cluster, we used a simple model expecting each shared cluster to be already present in the last common ancestor of the respective species. This approach is equivalent to inferring ancestral character states with Dollo parsimony (Farris, 1977).

### Comparing human genes with MXEs to orthologous genes in *Drosophila melanogaster*

Orthologous genes in *D. melanogaster* for all human genes containing MXE candidates were obtained with the Ensemble BioMart service (Yates *et al*, 2016). This list of orthologous genes was filtered with the list of *D. melanogaster* genes containing MXEs, which was obtained from Hatje and Kollmar (2013), to obtain a list of genes with both types of exons, (i) MXEs in human and MXEs in *D. melanogaster*, and (ii) MXE candidates in human but validated to be spliced differently and MXEs in *D. melanogaster*. Several of the human and *D. melanogaster* genes contain multiple clusters of MXEs. Thus, we compared all genes manually to determine whether MXEs are orthologous in both species, whether MXEs in human have orthologous exons in *D. melanogaster*, and whether MXEs in human do not correspond to exons in *D. melanogaster* genes.

### Data availability

All generated data can be searched, filtered and browsed at Kassiopeia (www.motorprotein.de/kassiopeia; Hatje & Kollmar, 2014). The primary RNA-Seq datasets used in this study are available in the following databases:

http://www.ebi.ac.uk/ena/data/view/ERP003613
http://www.ebi.ac.uk/ena/data/view/ERP000546
http://www.ncbi.nlm.nih.gov/geo/query/acc.cgi?acc = GSE36552
http://www.ncbi.nlm.nih.gov/geo/query/acc.cgi?acc = GSE44183
http://www.ncbi.nlm.nih.gov/geo/query/acc.cgi?acc = GSE33480
http://www.ncbi.nlm.nih.gov/geo/query/acc.cgi?acc = GSE30567

**Expanded View** for this article is available online.

### Acknowledgements

We would like to thank Prof. Paul Lingor and Lucas Araujo Caldi Gomes of the University Medicine Göttingen for providing total RNA isolated from human brain. We would like to thank Daniel Sumner Magruder from the Bonn group for critical suggestions. In particular, we would like to thank André Ahrens from the Kollmar group for his tremendous help in implementing the Allora database and performing the mapping of MXEs onto protein structures. The Kollmar group would like to thank Prof. Christian Griesinger for his continuous generous support. We would like to thank Dr. Robert P. Zinzen and Dr. Carla Margulies for critical reading of the manuscript.

### Author contributions

MK initiated the study and designed the analyses together with SB. KH and BH performed MXE predictions. KH integrated the human MXE data into the Kassiopeia database. KH implemented MXE candidate extraction and RNA-Seq analysis with help from ROV and SB. KH and MK did the cluster distribution, splicing mechanism and protein structure mapping analyses. KH, ROV, R-UR, VB, SB and MK performed the differential expression analysis. AR, MEM and TS performed the RT–PCR experiments. ROV, R-UR, VB and SB performed the SNP

mapping and prediction analysis. The comparison of different human gene annotations was done by KH and DS. KH did the prediction of MXEs in other mammals, and their comparison together with MK. MK and SB wrote the manuscript. ROV, KH, VB and R-UR contributed to manuscript text and Supplementary Materials. All authors read and approved the final manuscript.

## Conflict of interest

The authors declare that they have no conflict of interest.

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
