## [Review Process File · Molecular Systems Biology]

The landscape of human mutually exclusive splicing

Klas Hatje, Raza-Ur Rahman, Ramon O. Vidal, Dominic Simm, Björn Hammesfahr, Vikas Bansal, Ashish Rajput, Michel Edwar Mickael, Ting Sun, Stefan Bonn & Martin Kollmar

Review timeline:

Submission date:	2 May 2017
Editorial Decision:	29 May 2017
Revision received:	24 August 2017
Editorial Decision:	18 October 2017
Revision received:	4 November 2017
Accepted:	10 November 2017

Editor: Thomas Lemberger

Transaction Report:

1st Editorial Decision

29 May 2017

Thank you again for submitting your work to Molecular Systems Biology. We have now heard back from the three referees who agreed to evaluate your manuscript. As you will see from the reports below, the referees find the topic of your study of potential interest. They raise, however, a series of concerns which should be convincingly addressed in a major revision of this work.

Without repeating all the points raised by the reviewers, the major concerns refer to the following issues:

- a more detailed analysis is required to compare the global features observed on the set of mutually exclusive spliced genes and other forms of splicing. This includes functional enrichments, but also very importantly, the enrichment in pathogenic mutations and reported developmental stage specificities.
- given the existing Drosophila dataset, a human-Drosophila comparison is important
- reviewer #3 asks to include in the analysis the % of MEX inclusion (ϕ) as a more biologically relevant metric.
- reviewer #3 raises further points (points #3 and #4) with regard to experimental validation and refine the criterion for inclusion, to control and mitigate inclusion of spurious events.

We realize that reviewer #2 raises the issue of mechanistic insights. While the claims should be removed or strongly toned down, we do not feel that addressing this point experimentally would be necessary within the scope of this revision.

 REVIEWER REPORTS

Reviewer #1:

This manuscript reports the use of publicly available RNA-seq data, encompassing several tissues and developmental stages from different studies, to annotate and profile human mutually exclusive exons (MXEs) at the genome-wide level, aiming to study their mechanisms of regulation, physiological relevance and evolutionary dynamics.

For the purpose, the authors have devised a computational pipeline for the prediction and validation of MXEs, based on annotated and "de novo" predicted (from genomic alignments of RNA-seq data) protein-coding exons and splice junction read evidence. Information on reading frame, splice site and branch point sequence consensus, and RNA secondary structures was used to classify putative mechanisms of MXE regulation. Exon expression quantification and inequality metrics were used in differential MXE inclusion analysis of RNA-seq data and cross-species protein orthology searches were conducted in the evolutionary study of MXEs in mammals.

A main result of this work is a ~5-fold expansion of the repertoire of annotated MXEs, even when using very stringent criteria for their definition, suggesting that this type of alternative splicing is more common than previously thought. It is also observed that MXEs do not only occur in pairs, with clusters of up to ten MXEs being reported. The reported evolutionary analyses on MXEs provide evidence that events of exon duplication have accompanied speciation, consistently with previous reports showing that alternative splicing evolved rapidly, contributing to phenotypic diversity across species. Also, the tissue-specificity of the expression of MXEs highlights their role in generating biological diversity. Moreover, the association between the presence of pathogenic SNPs and patterns of MXE expression is argued to provide predictive clues for the global impact splicing has on disease traits.

Overall, we find the work well conceived and methodologically sound. The manuscript is well written and all the analyses are clearly explained, with their underlying assumptions reasonable and well justified. This work's stated main results are graphically illustrated in a comprehensive way. In general, the drawn conclusions are solidly supported by the presented results.

To our knowledge, this is the first reported global analysis of mutually exclusive splicing across human tissues and developmental stages, providing an unprecedentedly rich validated annotation of human MXEs and bringing new knowledge about the role of this mode of alternative splicing in our species and unveiling important features like the existence of multiple MXE clusters or the prevalence of MXEs integrating structurally constrained protein regions. It sets a "gold standard" set of stringent criteria for the definition of clusters of MXEs. It also innovates in cleverly applying inequality metrics (the Gini coefficient) in the context of transcriptomic data analysis.

This work makes a potentially relevant contribution to the alternative splicing research community, in particular to researchers working in human biology, creating an opportunity for new discoveries on splice variant functions and their impact in disease.

Our major concern about this manuscript is the absence of a true comparison between MXEs and other types of alternative splicing events. This weakness undermines the relevance of certain observations and, without such comparison, the findings in this manuscript will be deemed as of limited relevance and somewhat uncontextualized by the alternative splicing field.

For instance, in the 2nd paragraph of the Results section on ubiquitous and regulated exons (top of page 11), it is remarked that "many MXEs are specific to certain developmental stages". Isn't that the case for other types of alternative splicing? Are the genes harbouring those MXEs important in development?

Similarly, the major conclusions related to the higher susceptibility of MXEs to pathogenic mutations should include the comparison with other types of events, to further establish if the enrichment is in terms of mutually exclusive splicing or in splicing-associated sites in general. The discussion on the mammalian evolution of MXEs is also kept ambiguous, not clarifying what is special in the evolutionary dynamics of MXEs when compared to that of "cassette" exons, for instance.

Minor concerns:

1. The first sentence of the second paragraph of the Introduction should be carefully re-phrased to more accurately reflect what is stated in the cited literature. Is it true that "vertebrate MXEs have

been reported to only occur in pairs" or just that the cited papers only report MXEs in pairs but do not actually exclude the possible existence of clusters of more than two exons? Absence of evidence is not evidence of absence. This is actually better phrased in the penultimate paragraph in page 7: "there is to date no evidence of multi-cluster or higher order MXE clusters".

2. In the end of the first Results section (page 7), "the existence of further human MXEs" is suggested, as is, at the top of page 16, a "potential two-fold increase of the MXE-ome with further sequence data incorporated". It is not explained how this estimate was attained but Figure 1C suggests that it can even be made more accurate.
3. 5th line of page 9: both commas should be removed from "[...] all MXEs, whose structures have been analysed, are embedded [...]". As it is, the subject is all MXEs and not the MXEs whose structures have been analysed.
4. Page 9: the "three parameters that distinguish MXEs and cassette exons at a structural level" can be more specifically and clearly defined (e.g. the meaning of "high sequence similarity" and "far apart") and a reference to the respective section(s) in Materials and Methods should be made. Also, the use of "=>" can be misleading, given its ambiguous meaning.
5. Page 12: misplaced commas in "Interestingly, the percentage of genes with MXEs, which carry pathogenic SNPs, of all genes with MXEs is two-fold higher than the percentage of pathogenic SNP-carrying genes of all other genes". We suggest something like: "Interestingly, the percentage of pathogenic SNP-carrying genes with MXEs of all genes with MXEs is two-fold higher than the percentage of pathogenic SNP-carrying genes of all other genes".
6. The second paragraph of the Results section on pathogenic mutations (page 12) needs to be re-phrased in an unbiased way. The authors label as "surprising" the observation that "only" four SNP-containing MXE clusters contain mutations in both MXEs but do not quantitatively support their surprise by committing with an expected number. Similarly, the observation that many SNP-carrying MXEs are highly expressed can be just a consequence of an enrichment of the databases in SNPs on highly expressed (and therefore highly studied) areas of the genome.
7. The second part of the first paragraph of discussion (page 16) can now be enriched with references to TIBS letters with PMIDs 28483376 and 28483377.
8. Last paragraph of the Materials & Methods section on "Definition of criteria for RNA-seq evaluation of the MXE candidates": the statement on lower coverage of every single MXE than that of constitutive exons applies to other types of alternative splicing events. This should be mentioned, not to induce the reader to believe this an MXE-exclusive issue.
9. Discussion, bottom of page 16: it is either "40 times fewer datasets" or "40 times less data".
10. Legend of Figure 1F, page 39: the figure does not illustrate what the mentioned three levels of regulation are and they are not mentioned in the legend either. The meaning of the asterisks in the figure is also unclear.
11. Legend of Figure 4C, page 41: "[...] whereas the ROW [column] bar graph shows this for each tissue, cell type, and developmental stage."
12. Legend/labeling of Figure 5: the universe of exons represented in this figure needs to be made explicit. Besides, the authors need to clarify that Ensembl 37.75 "exons" are MXEs. Finally, they also need to explain what exons with no coloured bars for human are.
13. Y-axes of Figures 1D and 1E: "Number of" or "#" missing. The criteria for labeling axes should be made consistent across all main and supplementary figures.
14. Figures 3 and 4 and associated Supplementary Figures: the same colour scheme and order for the labeling of the 3 datasets (ENCODE, HPA, ED) should be used across all figures.

Reviewer #2:

Mutually exclusive splicing is an interesting and strictly regulated form of alternative splicing. In 2013, the lab described expansion of the mutually exclusive spliced exome in *Drosophila*. In this study, they described expansion of the mutually exclusive spliced exome in human, using similar method. This study shows the expression of over 855 MXEs, 42% of which represent novel exons. The data provides strong evidence for the existence of large and multi-cluster MXEs in higher vertebrates. These studies will help to aid our understanding of MXE evolution. However, the studies really do not offer new insights into MXE splicing mechanics and evolution as claimed in the Abstract, nor do their spatio-temporal expression predicts human disease pathology.

Major points

1. In abstract, the authors claimed "The data provides strong evidence for the existence of large and

multi-cluster MXEs in higher vertebrates and offers new insights into MXE splicing mechanics and evolution." However, I have not seen any new insights into MXE splicing mechanics in this study. Several mechanisms have been identified that serve to guarantee that pairs of alternative exons are spliced in a mutually exclusive manner, including steric interference between splice sites, spliceosome incompatibility and nonsense-mediated decay, competing RNA secondary structures. The former three mechanisms can explain how the alternative splicing of genes containing two mutually exclusive exons, but can not explain how the alternative splicing of genes containing more than two mutually exclusive exons (Graveley, Cell, 2005). Up to now, only competing RNA secondary structures can reasonably explain mutually exclusive splicing of more than two variable exons. The studies certainly do not reveal a novel mechanism of mutually exclusive splicing.

2. In abstract and the result section, the author claimed that MXEs are significantly enriched in pathogenic mutations. However, I have not seen any strong evidence, except for some examples and references. This part is highly speculative. Did this compare with constitutive exon, or with other alternatively spliced exon? Is the difference of statistical significance? Since both MXE-ratio expression and disease pathology are very complex, based on these results available in this study, I do not think MXE-ratio expression could predict disease pathology.

3. Page 8. "Mutually exclusive splicing is tightly regulated at the RNA and protein level". I could not really understand what the authors mean. Generally, alternative splicing (including mutually exclusive splicing) may be regulated by multiple cis-elements (including linear and structural elements, steric hindrance) and several trans-acting proteins. This part is confusing.

4. Since author lab ever published *Drosophila* mutually exclusive exome (Hatje & Kollmar, 2013), it is of interest to compare these two sets of data in human and *Drosophila*, including the ratio of mutually exclusive splicing, underlying mechanism, ect. How many are the overlap of MXEs in orthologous genes of human and *Drosophila*? How many are convergent evolutionary cases of MXEs in orthologous genes of human and *Drosophila*?

5. Page 14, "Rapid gain and loss of MXEs in mammalian evolution." is not clearly clarified. Actually, many MXEs are highly conserved across mammals. "Evolutionary dynamics of MXEs in mammalian" is more reasonable.

6. How are functional classifications for genes with MXEs?

Minor points

Fig S18. "The splicing of the exon10 cluster might be regulated by competing RNA secondary structure elements found in the introns following the MXEs and matching a selector element found in the intron after the last MXE." "a selector element " should be "the docking site"? How are these sequences conserved?

Reviewer #3:

In this manuscript, Hatje and co-authors present the first comprehensive analysis of mutually exclusive exon skipping (MXE) in humans. They used several features and billions of RNA-seq reads to predict and quantify MXEs, identifying hundreds of potential novel cases. Furthermore, they investigate their mechanisms of regulation, protein impact, association with disease-related SNPs, and evolution.

Overall, I think this is an important and timely study. However, I have a few methodological criticisms. I would be happy to support acceptance if these are properly addressed.

1) Relative isoform expression level is not an adequate measure to study the regulation of specific alternative splicing events. The authors should use percent of MEX inclusion (often referred to as percentage spliced in or psi) to investigate how the different isoforms are regulated, as the interpretation of relative expression levels is confounded by the overall differences in gene expression across tissues. The authors have already obtained exon-exon junction reads for all MXEs and conditions, so deriving psi's for each exon should not be difficult. With this metric, the authors will be able to better evaluate how predominant the major MXE is (is it included in 90% of the transcripts? 99%? This is a key point) and potentially detect tissue-dependent regulatory changes independent of changes in gene expression.

2) Relatedly, the authors should also use psi's to validate and assess the potential importance of novel MXEs. With such large amounts of RNA-seq data it is possible that some very lowly (perhaps spuriously) included MXEs are found. However, the biological relevance of a MXE included in <1% of the transcripts is unclear. Therefore, the authors should also report how many of their MXEs are present in <1%, <5%, etc. of the transcripts in all tissues.

3) Ideally, RT-PCR validations should be performed for a handful of novel MXE candidates in a few tissues, as cross-validation with RNA-seq data is sometimes not meaningful (e.g. if there is an intrinsic mapping issue in any of the intervening exon-exon junctions). Also, it would be good to see if the MXEs that are annotated as "constitutive exon pairs" (Type III, if I understood correctly) are in fact MXEs or could result from an RNA-seq mapping issue.

4) I am not in favor of considering pairs of exons with reads in the junctions between the alternative exons MXEs if the inclusion of both exons together causes a frame shift (criterion B in Methods). Firstly, the interest of MXEs comes from their mutual exclusion nature at the transcript level (most often due to exquisite splicing regulation). Secondly, most spurious alternative exons will create non-productive isoforms when included, so this may potentially end up with pairs of "proper" cassette exon + "spurious" cassette exon being defined as MXEs. I tried to find how many such MXEs not supported at the transcript level the authors found, but I could not find it. I suggest removing them from the analysis (e.g. if the number of reads between the alternative exons is >10% of those connecting any of the alternative exons to the constitutive exons), or at the very least treat them in a very different manner throughout. The claim that this study expands the catalog of MXEs by an order of magnitude may be an overstatement if previous catalogs have only considered proper MXEs.

Minor comments:

5) Page 10: the protein structure analysis is not very clear, and it does not look very informative in the present form. Also, Table S3 is very hard to digest. A more visual summary could be provided.

6) I found some calls to supplementary figures a bit misleading. For instance, S18, S19 and S29 do not provide much evidence for the claims made in the main text (at best, they provide a few illustrative examples).

7) Page 10: the authors should define in this section what "differentially expressed" means. As mentioned above, however, they should better use "differentially spliced", defined by the change in percent inclusion of the MXEs.

Reviewer #1 (Remarks to the Author):

Overall, we find the work well conceived and methodologically sound. The manuscript is well written and all the analyses are clearly explained, with their underlying assumptions reasonable and well justified. This work's stated main results are graphically illustrated in a comprehensive way. In general, the drawn conclusions are solidly supported by the presented results.

To our knowledge, this is the first reported global analysis of mutually exclusive splicing across human tissues and developmental stages, providing an unprecedentedly rich validated annotation of human MXEs and bringing new knowledge about the role of this mode of alternative splicing in our species and unveiling important features like the existence of multiple MXE clusters or the prevalence of MXEs integrating structurally constrained protein regions. It sets a "gold standard" set of stringent criteria for the definition of clusters of MXEs. It also innovates in cleverly applying inequality metrics (the Gini coefficient) in the context of transcriptomic data analysis.

This work makes a potentially relevant contribution to the alternative splicing research community, in particular to researchers working in human biology, creating an opportunity for new discoveries on splice variant functions and their impact in disease.

The reviewer raises a couple of concerns, all of which are well founded. We have addressed all points carefully and hopefully to the reviewers' expectations. In the end, we truly believe that the reviewer's comments and suggestions have raised the scientific impact of our study considerably.

Major concerns:

1. Our major concern about this manuscript is the absence of a true comparison between MXEs and other types of alternative splicing events. This weakness undermines the relevance of certain observations and, without such comparisons, the findings in this manuscript will be deemed as of limited relevance and somewhat un-contextualized by the alternative splicing field. For instance, in the 2nd paragraph of the Results section on ubiquitous and regulated exons (top of page 11), it is remarked that "many MXEs are specific to certain developmental stages". Isn't that the case for other types of alternative splicing? Are the genes harbouring those MXEs important in development?

This is actually an excellent suggestion and a very valid point of critique. A comparison to other forms of splicing would bring the

MXE results into the appropriate biological context while making the manuscript more accessible to the alternative splicing field. We have therefore analyzed and compared MXEs to cassette exons for their i) ontological enrichment, ii) development- and tissue-specific expression, and iii) enrichment in pathogenic SNPs. We have chosen to specifically look at cassette exons as they are similarly to MXEs often differentially spliced into transcripts.

- i. The GO analysis using WebGestalt showed that enriched terms between MXEs and cassette exons are largely different. While MXEs show a strong enrichment for muscle- and heart-related terms, cassette exons show enrichment for genes involved in organelle localization and microtubule function and development. In consequence, many of the MXE-enriched genes were transmembrane receptors or ion channels. We have adapted the main text ‘Many of the 1399 (855) MXEs have roles in the cardiac and muscle function and development, while cassette exons are enriched for microtubule- and organelle localization-related terms (Supplementary Fig. S14).’, the Materials and Methods, and have added Supplementary Fig. S14.
- ii. To compare the extent of tissue-specific expression of MXEs and cassette exons we calculated Gini indices for 1116 MXE pairs and compared the heterogeneity of expression to Gini indices of 4364 cassette exons (we did not have read mapping information for all cassette exons, only for the ones in genes containing potential MXE candidates). As can be seen in Supplementary Fig. S23, annotated MXEs and cassette exons are relatively homogeneously expressed across tissues, while especially novel MXEs show extremely high Gini indices, indicating selective tissue-specific expression. These results are in accordance with an excellent article by the Salzberg group, showing that especially novel cassette exons tend to have a more tissue-specific expression than annotated ones (Florea *et al.* 2013 ‘Thousands of exon skipping events differentiate among splicing patterns in sixteen human tissues’). We have included these results in the revised main text, changing ‘..., highlighting that MXE expression might be considerably more tissue-specific than the expression of other alternatively spliced exons.’, to ‘Interestingly, the expression of novel MXEs seems to be considerably more tissue-specific than the expression of annotated MXEs and cassette exons (Supplementary Fig. S23).’ and in novel Supplementary Fig. S23.

We would like to note that we have performed these analyses using Gini indices of percent-sliced-in (PSI) values (and regular counts), as suggested by reviewer 3.

- iii. For the analysis of pathogenic SNP enrichment please see the answer to major concern 2 (below).

Further comparison between mutually-exclusive and cassette exon splicing are made in the ‘Mutually exclusive presence of coding exons in functionally active transcripts’ section of the results (e.g. ‘In contrast to cassette exons and micro-exons, which tend to be located in surface loops and intrinsically disordered regions instead of folded domains (Buljan *et al*, 2012; Ellis *et al*, 2012; Irimia *et al*, 2014), all MXEs whose structures have been analysed are embedded within folded structural domains as has been shown for e.g. *DSCAM* (Meijers *et al*, 2007), *H2AFY* (Abascal *et al*, 2015a), the myosin motor domain (Kollmar & Hatje, 2014), and *SLC25A3* (Tress *et al*, 2016).’). Lastly, a brief comparison of MXEs and cassette exons can be found in the discussion ‘However, ribosome profiling data showed high frequencies of ribosome engagement of cassette exons indicating that these isoforms are likely translated (Weatheritt *et al*, 2016).’.

We are thankful for this excellent comment and hope that we have addressed it to the reviewer’s satisfaction.

2. Similarly, the major conclusions related to the higher susceptibility of MXEs to pathogenic mutations should include the comparison with other types of events, to further establish if the enrichment is in terms of mutually exclusive splicing or in splicing-associated sites in general.

Excellent suggestion: Next to the enrichment analysis of pathogenic SNPs in MXEs we have now also included an analysis of enrichment for cassette exons in the manuscript ‘Interestingly, the percentage of pathogenic SNP-carrying MXEs is two-fold higher than the percentage of all pathogenic SNP-carrying exons (Fisher’s exact test, p-value = $3 \cdot 10^{-11}$). A similar enrichment can be found for cassette exons (Fisher’s exact test, p-value = $2.2 \cdot 10^{-16}$) suggesting that in general alternative splicing-associated exons are susceptibility loci for pathogenic mutations.’ and the Materials and Methods ‘To access the statistical significance of disease enrichment in MXEs and cassette exons we compared the amount of pathogenic SNP-containing to non-SNP-containing exons. Out of 615410 annotated exons 21030 (3.4%) contain pathogenic SNPs, out of 1399 MXEs 99 (7.1%) contain pathogenic SNPs, and out of 31745 cassette exons 2143 (6.8%) contain pathogenic SNPs. The ~2-fold enrichment of

alternative splicing-associated exons (MXEs and cassette exons) is highly significant (Fisher's exact test, p-value MXE = $3 * 10^{-11}$, p-value cassette = $2.2 * 10^{-16}$).

In full congruence with the reviewer's expectation, it really seems that the enrichment is more in terms of splicing-associated sites / alternatively spliced exons in general. These findings further corroborate that some defining characteristics of MXEs and cassette exons are similar, including their splicing entropy and their enrichment in pathogenic SNPs (see also major point 1).

Again, we would like to thank the reviewer as the comparison of MXEs to other alternative splicing events truly raised the impact of this manuscript.

3. The discussion on the mammalian evolution of MXEs is also kept ambiguous, not clarifying what is special in the evolutionary dynamics of MXEs when compared to that of "cassette" exons, for instance.

We thank the reviewer for pointing out these ambiguities. We have completely revised this section, including now a detailed comparison of the human and the *Drosophila* MXEs as suggested by reviewer-3 (see also our comments to their requests). Merkin et al, 2012 and Barbosa-Morais et al, 2012 reported a set of about 500 cassette exons conserved in mammals, which seems to be a low number compared to the thousands of cassette exons identified in total. In contrast, the core set of MXEs present in eutherians and mammals, which would correspond to the evolutionary depth in Merkin et al and Barbosa-Morais et al., respectively, contains about 600 (350 in all mammals) MXEs, but the total number of MXEs is currently much lower compared to cassette exons. This supports the previously noted higher conservation of MXEs compared to cassette exons, which was put forward by just a few cases. Here, we present strong support at a genome scale. We hope, that the revised section better presents these findings now.

Minor concerns:

4. The first sentence of the second paragraph of the Introduction should be carefully re-phrased to more accurately reflect what is stated in the cited literature. Is it true that "vertebrate MXEs have been reported to only occur in pairs" or just that the cited papers only report MXEs in pairs but do not actually exclude the possible existence of clusters of more than two exons? Absence of evidence is not evidence of absence. This is actually better phrased in the

penultimate paragraph in page 7: "there is to date no evidence of multi-cluster or higher order MXE clusters".

The reviewer is absolutely correct and we have changed the text accordingly, now stating "Opposed to arthropods, current evidence suggests that vertebrate MXEs only occur in pairs ...".

5. In the end of the first Results section (page 7), "the existence of further human MXEs" is suggested, as is, at the top of page 16, a "potential two-fold increase of the MXE-ome with further sequence data incorporated". It is not explained how this estimate was attained but Figure 1C suggests that it can even be made more accurate.

This is indeed an excellent suggestion. We fit the sub-sampling data (x) to the number of expected MXEs $f(x)$ using Matlab and the optimal fit was obtained for a power function

$$f(x) = a \cdot x^b + c$$

where $f(x)$ is the expected number of MXEs, x is the number of reads, a is the linear coefficient, b is the exponential coefficient, and c is the error term. Although the fit is reasonably well, we do think that an extrapolation to potential saturation (number of expected MXEs in the genome) is not warranted (see 95% CI and deviation from 100% reads in the novel Suppl. Fig. S11B and below). We have therefore refrained from making any quantitative statements about the expected number of MXEs in the manuscript that go beyond a 2-fold increase in data.

Figure legend: To estimate the potential increase in MXEs given more sequencing data, we have fit the sub-sampling data to the number of expected MXEs $f(x)$ using Matlab. The green lines show the optimal fit for the expected number of validated MXEs in relation to the percentage of total RNA-seq reads used for validation (dark green 3 SJs 1 read; light green 3 SJs 3 reads). The actual measured data points are highlighted as yellow asterisks. The orange lines show the optimal fit for the expected number of initially ‘validated MXEs’ that will be rejected with increasing amounts of reads (dark orange 3 SJs 1 read; light orange 3 SJs 3 reads). The actual measured data points are highlighted as dark asterisks. Grey dashed lines indicate the predicted number of MXEs using 50, 100, 150, or 200% of the data (numbers are highlighted in the corresponding colors). Given a two-fold increase in the number of reads (100% – 200%), the expected number of validated MXEs (1SJ) is 1769 +/- 47 (95% confidence interval), validated MXEs (3SJ) is 1081 +/- 12, rejected MXEs (1SJ) is 227 +/- 9, and the number of rejected MXEs (3SJ) is 95 +/- 5. While the number of validated MXEs is far from saturation (a 100% increase in data results in 27% increase in the number of validations) the number of rejected MXEs seems to be saturated (a 100% increase in data results in 2% increase in the number of rejections).

In consequence, we have made many changes to the material and methods section ‘Saturation analysis’. In addition, we have included a reference to the ‘Saturation analysis’ section in the main text ‘To estimate the dependence of MXE confirmation and rejection on data

quantity we cross-validated the MXE gain (validation) and loss (rejection) events for several subsets of the total RNA-seq data (Fig. 1C, Supplementary Fig. S11, Materials and Methods ‘Saturation analysis’). We have included a novel Suppl. Fig. S11B that highlights the results of the extrapolation. In the discussion, we have changed the sentence ‘Saturation analysis and the existence of 1816 expressed but unconfirmed MXE candidates promise a potential two-fold increase with further sequence data incorporated.’ to ‘Saturation analysis and the existence of 1816 expressed but unconfirmed MXE candidates suggest a potential 27% increase of the MXE-ome with a two-fold increase in data.’

6. 5th line of page 9: both commas should be removed from "[...] all MXEs, whose structures have been analysed, are embedded [...]". As it is, the subject is all MXEs and not the MXEs whose structures have been analysed.

We have removed the commas in accordance with the reviewers comment.

7. Page 9: the "three parameters that distinguish MXEs and cassette exons at a structural level" can be more specifically and clearly defined (e.g. the meaning of "high sequence similarity" and "far apart") and a reference to the respective section(s) in Materials and Methods should be made. Also, the use of " \Rightarrow " can be misleading, given its ambiguous meaning.

We thank the reviewer for the suggestion to revise this section. This was also suggested by the other reviewers, and we hope that the revised section is better understandable now. To illustrate the concept we have added an illustration of the protein structure of macroH2A1 (*H2AFY* gene) to Supplementary Fig. S19 (now S22). In this figure, we marked the region encoded by the MXE, specifically marked the exon ends, which are located within secondary structural elements, and the distance between these ends. We chose *H2AFY* as example, because this was already marked in Fig. 2C and it is an example for low sequence similarity and short distance, as compared to other MXEs. We hope, that this example can serve the reader as guidance for judging other cases presented in Fig. 2C. We do not give numbers for “high sequence similarity” and “far apart” because those might be misinterpreted as cut-offs. However, every cluster of MXEs needs to be analyzed and interpreted with respect to all parameters. For example, many of the MXE candidates that we validated to be spliced constitutively, also show “high sequence similarity” and might present structural repeat regions. However, if

e.g. the N-terminal end of the exon-encoded peptides is part of a long alpha helix and the C-terminus ends within an extended loop, it seems highly unlikely that another exon with high similarity (suggesting similar spatial interactions of the encoded peptide) could be joined to this C-terminus and start within a loop. We hope that by revising the entire section our intentions to provide a structural basis to distinguish MXEs from cassette exons are clearer now. We thank the reviewer for pointing out the ambiguous meaning of “=>” and have removed this accordingly.

8. Page 12: misplaced commas in "Interestingly, the percentage of genes with MXEs, which carry pathogenic SNPs, of all genes with MXEs is two-fold higher than the percentage of pathogenic SNP-carrying genes of all other genes". We suggest something like: "Interestingly, the percentage of pathogenic SNP-carrying genes with MXEs of all genes with MXEs is two-fold higher than the percentage of pathogenic SNP-carrying genes of all other genes".

We have changed the text in accordance with the reviewer's suggestion (verbatim).

9. The second paragraph of the Results section on pathogenic mutations (page 12) needs to be re-phrased in an unbiased way. The authors label as "surprising" the observation that "only" four SNP-containing MXE clusters contain mutations in both MXEs but do not quantitatively support their surprise by committing with an expected number. Similarly, the observation that many SNP-carrying MXEs are highly expressed can be just a consequence of an enrichment of the databases in SNPs on highly expressed (and therefore highly studied) areas of the genome.

Indeed we have no reason to believe that this is a surprising finding, since we have not performed a statistical analysis. We have therefore removed any qualitative statement from the sentence 'Four of all SNP-containing MXE clusters contain mutations in both MXEs (*FHL1*, *MAPT*, *CACNA1C* and *CACNA1D*), whereas 31 currently have pathogenic SNPs in only one MXE.'

10. The second part of the first paragraph of discussion (page 16) can now be enriched with references to TIBS letters with PMIDs 28483376 and 28483377.

We are happy to include these two TIBS letters in the references as suggested.

11. Last paragraph of the Materials & Methods section on "Definition of criteria for RNA-seq evaluation of the MXE candidates": the statement on lower coverage of every single MXE than that of constitutive exons applies to other types of alternative splicing events. This should be mentioned, not to induce the reader to believe this an MXE-exclusive issue.

This is a very good observation and we have changed the sentence to 'Note that as a matter of principle the read coverage of MXEs and other alternative splicing events is considerably lower than that of constitutive exons due to their mutually exclusive inclusion in the transcripts.'

12. Discussion, bottom of page 16: it is either "40 times fewer datasets" or "40 times less data".

This is correct and we have changed the text to '40 times less data'.

13. Legend of Figure 1F, page 39: the figure does not illustrate what the mentioned three levels of regulation are and they are not mentioned in the legend either. The meaning of the asterisks in the figure is also unclear.

We apologize for our mistake, as neither the text nor the legend contained the relevant information. As we did not want to speculate too much on potential mechanisms of regulation we decided to restrict our description of *CUX1* to its clusters 'The *CUX1* gene (cut-like homeobox 1) contains 2 interleaved clusters of MXEs (clusters 1 and 2) and 2 standard clusters each with two MXEs (clusters 3 and 4). The exon 3 and exon 4 variants each are orthologous exons. The exon 4 variants are mutually exclusive (cluster 2). Exon 3a is a differentially included exon and only spliced together with exon 4a. The exons 3b, 3c, 3d and 3e are part of a cluster of four MXEs (cluster 1) and are only spliced together with exon 4b (Supplementary Fig. S16 and S17). Novel exons are labelled with an asterisk.'

14. Legend of Figure 4C, page 41: "[...] whereas the ROW [column] bar graph shows this for each tissue, cell type, and developmental stage."

Thank you for noticing this mistake. We have changed the legend to '...whereas the row bar graph shows this for each tissue...'

15. Legend/labeling of Figure 5: the universe of exons represented in this figure needs to be made explicit. Besides, the authors need to

MAX-PLANCK-GESELLSCHAFT

clarify that Ensembl 37.75 "exons" are MXEs. Finally, they also need to explain what exons with no coloured bars for human are.

We appreciate the reviewer for indicating inaccurate and misleading phrasing in the figure legend. We have revised the legend and hope that the figure is well understandable now.

16. Y-axes of Figures 1D and 1E: "Number of" or "#" missing. The criteria for labeling axes should be made consistent across all main and supplementary figures.

This is a valid point and we have added '#' to every axes that contains number information. In addition, we have tried our best to harmonize the figure labeling throughout the manuscript (see also minor concern 17).

17. Figures 3 and 4 and associated Supplementary Figures: the same colour scheme and order for the labeling of the 3 datasets (ENCODE, HPA, ED) should be used across all figures.

This is a very good suggestion and we harmonized the color scheme accordingly. The ordering of the 3 datasets, especially in the heatmaps, depends on the computed dendrogram and can therefore vary.

Reviewer #2 (Remarks to the Author):

Mutually exclusive splicing is an interesting and strictly regulated form of alternative splicing. In 2013, the lab described expansion of the mutually exclusive spliced exome in *Drosophila*. In this study, they described expansion of the mutually exclusive spliced exome in human, using similar method. This study shows the expression of over 855 MXEs, 42% of which represent novel exons. The data provides strong evidence for the existence of large and multi-cluster MXEs in higher vertebrates. These studies will help to aid our understanding of MXE evolution. However, the studies really do not offer new insights into MXE splicing mechanics and evolution as claimed in the Abstract, nor do their spatio-temporal expression predicts human disease pathology.

The reviewer raises several crucial concerns and we have tried to address all of them appropriately, which allowed us to significantly improve the quality of this manuscript. We would therefore like to thank the reviewer for raising the scientific accuracy and impact of our research.

Major concerns:

1. In abstract, the authors claimed "The data provides strong evidence for the existence of large and multi-cluster MXEs in higher vertebrates and offers new insights into MXE splicing mechanics and evolution." However, I have not seen any new insights into MXE splicing mechanics in this study. Several mechanisms have been identified that serve to guarantee that pairs of alternative exons are spliced in a mutually exclusive manner, including steric interference between splice sites, spliceosome incompatibility and nonsense-mediated decay, competing RNA secondary structures. The former three mechanisms can explain how the alternative splicing of genes containing two mutually exclusive exons, but cannot explain how the alternative splicing of genes containing more than two mutually exclusive exons (Graveley, Cell, 2005). Up to now, only competing RNA secondary structures can reasonably explain mutually exclusive splicing of more than two variable exons. The studies certainly do not reveal a novel mechanism of mutually exclusive splicing.

This is actually a very important point and we are really sorry if we suggested anywhere in the manuscript that we have found a novel mechanism of mutually exclusive splicing. The sentence 'The data provides strong evidence for the existence of large and multi-cluster MXEs in higher vertebrates and offers new insights into MXE

splicing mechanics and evolution.’ was meant to highlight that we have, for the first time, systematically analyzed the frequency of potential mechanisms across hundreds of MXEs. This led us to the conclusion that most MXE events seem to be regulated by RNA secondary structures, exactly as the reviewer has stated.

In consequence, we changed any text that would only remotely insinuate that we have analyzed or identified novel splicing mechanisms. In more detail, we have changed the abstract from ‘The data provides strong evidence for the existence of large and multi-cluster MXEs in higher vertebrates and offers new insights into MXE splicing mechanics and evolution.’ to ‘The data provides strong evidence for the existence of large and multi-cluster MXEs in higher vertebrates and offers new insights into MXE evolution.’. In the results section ‘Mutually exclusive splicing is tightly regulated at the RNA and protein level’ we have changed ‘To gain mechanistic insights into the regulation of mutually exclusive splicing in humans we investigated four mechanisms that were shown to act in some specific cases and were proposed to coordinate mutually exclusive splicing in general (Fig. 2A) (Letunic *et al*, 2002; Smith, 2005).’ to ‘To understand which splicing mechanisms might be primarily responsible for the regulation of mutually exclusive splicing in humans we investigated several mechanisms that were shown to act in some specific cases and were proposed to coordinate mutually exclusive splicing in general (Fig. 2A) (Letunic *et al*, 2002; Smith, 2005).’. There is no other part of the manuscript that discusses or highlights the ‘mechanistic analysis’ if we may call it so.

Again, we are greatly sorry for our mistake but we do hope that the revised document captures the novelty of the manuscript better.

2. In abstract and the result section, the author claimed that MXEs are significantly enriched in pathogenic mutations. However, I have not seen any strong evidence, except for some examples and references. This part is highly speculative. Did this compare with constitutive exon, or with other alternatively spliced exon? Is the difference of statistical significance? Since both MXE-ratio expression and disease pathology are very complex, based on these results available in this study, I do not think MXE-ratio expression could predict disease pathology.

The reviewer raises two concerns, i) are MXEs enriched for pathogenic mutations (also compared to other splice isoforms), ii) they do not believe (SNP-containing) MXE expression could predict pathogenesis. We answer these two points separately:

- i. First of all we would like to apologize because neither the

main text nor the Materials and Methods section gave a sufficient description of how the significant enrichment analysis was performed. We can therefore fully understand this point of critique. The revised manuscript now contains a detailed section describing the rationale and statistics used to assess pathogenic SNP enrichment (Materials and Methods section ‘Identification of pathogenic SNPs in MXEs’), for MXEs and cassette exons (see point 2 of reviewer 1). In brief, we have now added the sentence ‘Interestingly, the percentage of pathogenic SNP-carrying MXEs is two-fold higher than the percentage of all pathogenic SNP-carrying exons (Fisher’s exact test, p-value = $3 \cdot 10^{-11}$). A similar enrichment can be found for cassette exons (Fisher’s exact test, p-value = $2.2 \cdot 10^{-16}$) suggesting that in general alternative splicing-associated exons are susceptibility loci for pathogenic mutations.’ to the main text and added the following description to the Materials and Methods ‘To assess the statistical significance of disease enrichment in MXEs and cassette exons we compared the amount of pathogenic SNP-containing to non-SNP-containing exons. Out of 615410 annotated exons 21030 (3.4%) contain pathogenic SNPs, out of 1399 MXEs 99 (7.1%) contain pathogenic SNPs, and out of 31745 cassette exons 2143 (6.8%) contain pathogenic SNPs. The ~2-fold enrichment of splicing-associated exons (MXEs and cassette exons) is highly significant (Fisher’s exact test, p-value MXE = $3 \cdot 10^{-11}$, p-value cassette = $2.2 \cdot 10^{-16}$).’.

We thank the reviewer for raising this important point and apologize for our lack of clarity in the initial manuscript.

- ii. We do agree with the reviewer that both, MXE-ratio expression and disease pathology are complex phenomena. Our data do not have the numbers to learn exact pathologies or give developmental time resolution and it is debatable if this can be achieved given much larger datasets and information.

To accommodate for our low number of observations we first abstracted the pathology to the organ it affects, as described in detail in the material and methods. We then used leave-one-out cross-validation to robustly predict the affected organ from the MXE-ratio expression.

In essence, our data give reasonable indication that exon expression, not gene expression per se, can predict pathology when overlaid with pathogenic SNP information. We agree with the reviewer, however, that our claims should be toned down to reflect our findings better. In consequence, we have

changed the abstract from ‘Finally, MXEs are significantly enriched in pathogenic mutations and their spatio-temporal expression predicts human disease pathology.’ to ‘Finally, MXEs are significantly enriched in pathogenic mutations and their spatio-temporal expression **might** predict human disease pathology.’ In all other parts of the manuscript we have used the conjunctive to describe our findings (‘Although based on only 24 observations, our data suggest that MXE expression might predict disease pathogenicity in space and potentially also in time.’ in the results section, ‘Furthermore, our data suggest that MXE expression might reflect disease pathogenesis that could allow for the prediction of the affected organ(s). It is intriguing to speculate that the observed expression-disease association is a general dogma, which could be used to predict yet unseen diseases from published expression data, potentially bringing about a paradigmatic shift in (computational) disease research.’ in the discussion).

In conclusion, we thank the reviewer for the excellent comment and hope that we have answered the above concerns sufficiently.

Addendum: We have extended the approach to predict pathology from MXE expression (in this manuscript) to all exons and the concept seems to generalize surprisingly well.

3. Page 8. "Mutually exclusive splicing is tightly regulated at the RNA and protein level". I could not really understand what the authors mean. Generally, alternative splicing (including mutually exclusive splicing) may be regulated by multiple cis-elements (including linear and structural elements, steric hindrance) and several trans-acting proteins. This part is confusing.

We have completely revised this section and hope that it is clearly understandable now. We have rephrased the heading to “Mutually exclusive presence of coding exons in functionally active transcripts”. Of course, generating transcripts that are subject to degradation by NMD is not a form of mutually exclusive splicing per se. However, as with many biological processes mutually exclusive splicing by e.g. competing RNA secondary structures or steric hindrance is not 100% perfect but results in higher or lower error rates, depending on each case. Thus, we would still regard a cluster of MXEs with 20000 supporting SJ reads for each site as tightly regulated if e.g. 10-20 MXE-joining reads were found. This is the case for most of the annotated MXEs, where we found >10 MXE-joining reads for 75% of the cases (91 of 122 annotated MXEs). This

is shown in detail in Supplementary Figs. S3A and S3D. Of the novel predicted exons, which are in general less supported by reads, we only find MXE-joining reads for 25 of 615 MXEs. We do not have a model to quantify splicing errors yet, which might be tissue specific, developmental stage specific, depend on pre-mRNA length and many more parameters. Thus, we cannot quantify the term “tightly regulated” yet. We have thoroughly revised this section with respect to the correct usage of all terms.

4. Since author lab also published the *Drosophila* mutually exclusive exome (Hatje & Kollmar, 2013), it is of interest to compare these two sets of data in human and *Drosophila*, including the ratio of mutually exclusive splicing, underlying mechanism, ect. What is the overlap of MXEs in orthologous genes of human and *Drosophila*? How many convergent evolutionary cases of MXEs in orthologous genes of human and *Drosophila* are there?

Initially we thought that such an analysis would go far beyond the scope of our manuscript. For example, convergent or divergent evolution cannot be proven by just analyzing human and *Drosophila*, this would need the analysis of far more species from sister clades (lophotrochozoans, nematodes, hemichordates, echinoderms, etc.) and metazoans that diverged before bilaterians, e.g. cnidarians, porifera and placozoans. We have done such a taxonomically broad analysis already for muscle myosin heavy chain genes (Kollmar and Hatje, PlosOne 2014). Performing similar analyses for all genes with MXEs in human and/or *Drosophila* would be very interesting but would also be far away from other aspects of our manuscript such as spatio-temporal expression patterns and disease relevance. In evolutionary terms, humans and *Drosophila* are just two single species of mammals and insects and a comprehensive analysis would need to also point out the “missing data”, MXEs that have been lost in humans/mammals and *Drosophila*. In our analysis of the myosin genes we could show that *Drosophila* has a very restricted set of MXEs compared to more intron-rich species such as mollusks and *Daphnia*. Currently, we do not even have any data on MXEs that have been lost in humans. In 2008, we did another analysis of arthropod muscle myosin heavy chain genes (Odriontz and Kollmar, BMC Mol Biol) including in-depth phylogenetic analyses of all MXEs. This demonstrated ancestry of both MXEs in some clusters, but very difficult to explain scenarios for other clusters, which seemed to include multiple independent exon duplication and loss events. The scenario of convergent evolution as presented in Trends in Genetics (2004) by R.Copley seems compelling, but R.Copley compared non-orthologous genes (e.g. human SCN genes with

Drosophila cac instead of para) and data from more taxa was not available.

Keeping these restrictions in mind, we have compared the genes containing MXEs in human and *Drosophila* as suggested, and the data is presented in multiple supplementary figures and a new supplementary table (Supplementary Table S10). This is, however, not an exhaustive analysis and we do not provide numbers for “ratio of mutually exclusive splicing”, because it is not clear which ratio would represent this in a meaningful way. Ratios could be comparison of total numbers of MXEs, MXE clusters, numbers of MXEs/MXE clusters per gene or transcript, or per total numbers of exons in the genome, etc. Also, we do not speculate about convergent/divergent evolution as this would need to include data from other bilaterians. We hope that our analysis is in the sense of the reviewer’s expectations although it is only a first glimpse on the evolution of MXEs in the context of bilaterians.

5. Page 14, "Rapid gain and loss of MXEs in mammalian evolution." is not clearly clarified. Actually, many MXEs are highly conserved across mammals. "Evolutionary dynamics of MXEs in mammalian" is more reasonable.

We absolutely agree with the reviewer and have changed the section header accordingly ‘Evolutionary dynamics of MXEs in mammals and bilaterians’. Bilaterians were added because of the now included comparison to the *Drosophila* MXEs as suggested above.

6. How are functional classifications for genes with MXEs?

This is actually an interesting question. An enrichment analysis using WebGestalt shows that MXEs are enriched for muscle and synapse-related terms, while cassette exons are enriched for the terms ‘microtubule’ and ‘organelle localization’. Many of the MXE-enriched terms are directly related to ion channels and transmembrane receptors, reflecting gene size, splicing susceptibility, and functional importance of mutually-exclusive splicing in these tissues.

Enrichment results can be found in the main text ‘Many of the 1399 (855) MXEs have roles in the cardiac and muscle function and development, while cassette exons are enriched for microtubule- and organelle localization-related terms (Supplementary Fig. S14).’, the Materials and Methods ‘We used WebGestalt for Gene Ontology enrichment analyses (Wang et al, 2013). The lists of unique genes in gene symbol format were uploaded to WebGestalt and the GO Enrichment Analysis selected. The entire human genome annotation

MAX-PLANCK-GESELLSCHAFT

was set as background and 0.05 as threshold for the p-value for the significance test using the default statistical method "hypergeometric". Categorical enrichment of MXEs and cassette exons was summarized in a heatmap.', and in new Suppl. Fig. S14.

Minor concerns:

7. Fig S18. "The splicing of the exon10 cluster might be regulated by competing RNA secondary structure elements found in the introns following the MXEs and matching a selector element found in the intron after the last MXE.". "a selector element " should be "the docking site"? How are these sequences conserved?

We appreciate the reviewer's careful reading of this figure's legend, and apologize for the wrong labeling of the docking site and the selector elements. This has been corrected in the figure and the legend. As pointed out in the legend, the MXE cluster is only conserved in human and chimpanzee.

Reviewer #3 (Remarks to the Author):

In this manuscript, Hatje and co-authors present the first comprehensive analysis of mutually exclusive exon skipping (MXE) in humans. They used several features and billions of RNA-seq reads to predict and quantify MXEs, identifying hundreds of potential novel cases. Furthermore, they investigate their mechanisms of regulation, protein impact, association with disease-related SNPs, and evolution.

Overall, I think this is an important and timely study. However, I have a few methodological criticisms. I would be happy to support acceptance if these are properly addressed.

The reviewer's suggestions and concerns are throughout well founded and have significantly improved the scientific quality of the manuscript. Especially the suggestion to use PSI values over other approaches was very help- and insightful. We have addressed all of the reviewer's comments and concerns in the revised manuscript.

Major concerns:

1. Relative isoform expression level is not an adequate measure to study the regulation of specific alternative splicing events. The authors should use percent of MEX inclusion (often referred to as percentage spliced in or psi) to investigate how the different isoforms are regulated, as the interpretation of relative expression levels is confounded by the overall differences in gene expression across tissues. The authors have already obtained exon-exon junction reads for all MXEs and conditions, so deriving psi's for each exon should not be difficult. With this metric, the authors will be able to better evaluate how predominant the major MXE is (is it included in 90% of the transcripts? 99%? This is a key point) and potentially detect tissue-dependent regulatory changes independent of changes in gene expression.

This is an extremely important point and we could not agree more. PSI values do not contain a potential gene expression bias, although we would like to highlight the fact that the statistical model in DEXSeq normalizes for gene expression differences when estimating differential exon inclusion.

We have now re-evaluated differential exon inclusion using PSI and delta PSI values in the sections 'MXEs mainly consist of one ubiquitous exon and otherwise regulated exons' and 'MXEs are high-susceptibility loci for pathogenic mutations' and adjusted the text, figures, and tables accordingly. In more detail, we have now

used splice-junction bridging reads to calculate PSI and delta PSI values and used a Kruskal-Wallis rank sum test and a BH multiple testing correction to estimate differential inclusion significance (see also revised Materials and Methods section).

The results that we have obtained using PSI values closely resemble the differential inclusion results with DEXSeq. Thus, we found 499 genes containing 914 differentially expressed MXEs (65% of the total 1399 MXEs) using count data and DEXSeq. Using a Kruskal-Wallis rank sum test with PSI values we found 519 genes containing 942 differentially expressed MXEs (67% of the total 1399 MXEs). Thus, 71% of the differentially included MXEs detected with DEXSeq were also detected using PSI values (69% of the PSI differentially included MXEs were also detected using DEXSeq).

Figure legend: Comparison of the differentially included MXEs using a count-based parametric approach (DEXSeq) and PSI values with a non-parametric test (Kruskal-Wallis rank sum test).

Furthermore, we have also used delta PSI values and PSI values to predict disease. The results obtained using PSI and RPKM-based machine learning are also very similar, reaching accuracies between 79% and 82%. These results are shown in the revised main text, Fig. 4 and Suppl. Fig. S29.

We believe that the inclusion of PSI value-based analyses has significantly strengthened the results and we would like to thank the reviewer for raising this excellent point.

2. Relatedly, the authors should also use psi's to validate and assess the potential importance of novel MXEs. With such large amounts of RNA-seq data it is possible that some very lowly (perhaps spuriously) included MXEs are found. However, the biological relevance of a MXE included in <1% of the transcripts is unclear. Therefore, the authors should also report how many of their MXEs are present in <1%, <5%, etc. of the transcripts in all tissues.

Also this comment is excellent as the MXEs that are hardly ever spliced into transcripts might have questionable biological relevance. As suggested, we therefore analyzed the PSI and delta PSI values of all MXE pairs that contain a novel predicted exon (Supplementary Fig. S23 C-D). Of all novel MXEs, less than 17% are included in less than 1% of the transcripts. Half of the novel MXEs is spliced in in over 5% of the transcripts, whereas 20% are spliced in in over 50% of the transcripts. We can therefore state with certainty that the vast majority of novel MXEs are spliced into transcripts with relatively high frequency.

We have included this analysis in the revised manuscript in Suppl. Fig. S23 C-D showing the PSI and delta PSI values for novel MXEs, annotated MXEs, and cassette exons (see comments of reviewer 1). In addition, we have amended the Material and Methods section to include the PSI-based analyses (see also the first point of critique).

3. Ideally, RT-PCR validations should be performed for a handful of novel MXE candidates in a few tissues, as cross-validation with RNA-seq data is sometimes not meaningful (e.g. if there is an intrinsic mapping issue in any of the intervening exon-exon junctions). Also, it would be good to see if the MXEs that are annotated as "constitutive exon pairs" (Type III, if I understood correctly) are in fact MXEs or could result from an RNA-seq mapping issue.

Mapping algorithms can cause artifacts on multiple levels and it is a very good suggestion of the reviewer to closely inspect if the bulk of the MXEs predicted are real MXEs. We have taken multiple steps to validate our data:

- i. qPCR validation: As suggested by the reviewer, we have selected 6 MXEs for validation with qPCR in brain tissue (unfortunately it is not easy for us to obtain any/other human material). In this list are also two MXEs that were annotated as 'cassette exon pairs' (ACSL6 and MEF2C). All assayed MXEs, with the exception of Rab35, showed perfect coherence with our mapping results, validating our analysis

algorithm on a small scale. For Rab35, we were not able to design a functional UP-MXE1 primer, all other results met expectations. These results are summarized in the novel Suppl. Table S3, the novel Suppl. Fig. S13, and the revised Materials and Methods section.

- ii. qPCR is good to validate few examples but not suited to obtain information on potential mapping issues in general (if e.g. only 10% of the MXEs are due to mapping bias). We have therefore compared MXEs with two annotated exons to splicing data in GTEx portal (<https://www.gtexportal.org/home/>). Again, we could validate all MXEs with annotated exons using data that was mapped with a different read aligner (GTEx uses bowtie 2, we used STAR) and very different parameters (especially for SJ mapping we used more stringent criteria to avoid mis-mapping). Also these results we have added to the revised manuscript to further strengthen the quality and impact of our results (Suppl. Fig. S12).

We also summarized the MXE validation in the main text of the manuscript ‘To further validate the list of MXEs, we compared MXEs that contained two ‘annotated other splicing’ exons to splicing information from GTEx portal (<https://www.gtexportal.org/home/>). Although GTEx portal uses an alternative aligner and alignment settings, all MXEs that we compared showed mutually exclusive behaviour in GTEx portal (Supplementary Fig. S12), substantiating our results. Lastly, we selected 6 brain-expressed novel MXEs for qPCR validation in human brain total RNA. All assayed MXEs showed perfect coherence with the alignment results, confirming mutually exclusive splicing of all assayed novel MXEs in human brain (Supplementary Fig. S13, Supplementary Table S3).’. In conclusion, we want to thank the reviewer for this very good suggestion and we believe that we could conclusively show that the detected MXEs are true mutually exclusive splicing events..

4. I am not in favor of considering pairs of exons with reads in the junctions between the alternative exons MXEs if the inclusion of both exons together causes a frame shift (criterion B in Methods). Firstly, the interest of MXEs comes from their mutual exclusion nature at the transcript level (most often due to exquisite splicing regulation). Secondly, most spurious alternative exons will create non-productive isoforms when included, so this may potentially end up with pairs of "proper" cassette exon + "spurious" cassette exon being defined as MXEs. I tried to find how many such MXEs not

supported at the transcript level the authors found, but I could not find it. I suggest removing them from the analysis (e.g. if the number of reads between the alternative exons is $>10\%$ of those connecting any of the alternative exons to the constitutive exons), or at the very least treat them in a very different manner throughout. The claim that this study expands the catalog of MXEs by an order of magnitude may be an overstatement if previous catalogs have only considered proper MXEs.

We hope the reviewer doesn't mind if we respectfully put forward a different point of view. Although 377 (60%) of the 629 MXE clusters contain exons with lengths not divisible by three thus leading to frame-shift in case of combined inclusion, such MXE-joining reads were only found for 83 of these clusters (13%). Notably, these 83 MXE clusters contain the majority of the annotated MXEs (91 of 122 MXEs), many of the exons that were previously annotated as other splice type (44 of 662 MXEs), but only few of the novel MXEs predicted in intronic regions (4% or 25 of 615 MXEs; Supplementary Fig. S3D). These cases of annotated MXEs with exon-joining reads include the well-known MXEs of the three tropomyosin genes, the *SCN1A*, *SCN2A*, *SCN8A*, *SCN9A* sodium channels, the *GLRA2* receptor gene, the *CACNIC* and *CACNID* calcium channel genes, and many more. Supplementary Fig. S3D also shows, that we find >10 exon-joining reads for most of these cases, indicating that these are not "spurious" exons. Otherwise, these would not have been annotated already as MXEs. If we excluded these types of MXEs (those with exon-joining reads) from the main dataset and treat them differently, we would very likely confuse the entire MXE-community, because most of the human MXE cases described in the literature would be missing in what we would need to describe as "proper MXE dataset".

We do not believe that a certain percentage of MXE-joining reads could be a good filter for including or excluding clusters, as suggested by the " $>10\%$ " by the reviewer. 10% is, to our knowledge, far above the error rate of the spliceosome. On the other hand, 150 joining reads would not indicate a spurious event (compared to the many cases of MXEs validated by 10-50 reads), if these would just represent 8-9% of the total MXE-validating reads and thus not be filtered out. In contrast, exon-joining reads might not only exist for exons not divisible by three but also for exons divisible by three. We are sure that we also excluded many true MXEs from the dataset because of this criterion. There might be many MXE candidates supported by $>10,000$ reads but rejected by just a single exon-joining read.

Finally, we would like to apologize that the above-mentioned aspects were not presented as clear as they should have been in the manuscript. The numbers the reviewer requests, for example, were presented in Supplementary Figs. S3A and S3D and referenced in the mechanistic insights section. In consequence, we revised large parts of the initial manuscript, especially the ‘mechanistic insights’ section, and tried to improve the wording and phrasing as far as possible.

Minor concerns:

5. Page 10: the protein structure analysis is not very clear, and it does not look very informative in the present form. Also, Table S3 is very hard to digest. A more visual summary could be provided.

We appreciate the reviewer’s careful reading of this section, and their reviewing of the supplementary material. Misunderstanding phrasing in this section has also been pointed out by the other reviewers. We have thoroughly revised the entire section about the splicing mechanism and the structural analysis, revised Table S3 (now Table S4), and added a scheme of the protein structure analysis process to the Supplementary Fig. S19 (now S22). Please see also our comments on similar requests by the other reviewers.

6. I found some calls to supplementary figures a bit misleading. For instance, S18, S19 and S29 do not provide much evidence for the claims made in the main text (at best, they provide a few illustrative examples).

We apologize that the links to Supplementary Figures S18 & S19 (now Figs. S21 and S22) were misleading. As response to minor request 5, we have considerably extended Fig. S19 and, in consequence, we have shifted the reference for S19, now stating ‘To assess this model, MXEs were mapped against the PDB database (Fig. 2C, Supplementary Fig. S22, Supplementary Table S4) (Rose *et al.*, 2015).’. We believe this is the correct place to link the figure. Supplementary Fig. S19 contained the distribution of PDB structures with mapped MXEs across organisms, which was also mentioned in the original manuscript text. Although we feel inclined to retain this figure in the manuscript, especially with the revised and augmented content, we would remove it if the reviewer feels it is disruptive or unimportant for this manuscript.

With respect to Supplementary Fig. S18, we have rephrased the part around its citation to “Competing RNA secondary structures are, however, usually not conserved across long evolutionary

MAX-PLANCK-GESELLSCHAFT

distances. A potential case of a docker site and selector sequences downstream of each exon variant was identified for the cluster of four MXEs in the *CD55* gene (Supplementary Fig. S21).” We believe that such an example might be helpful for the non-expert reader not aware of the concept of the competitive RNA secondary structural elements. As the mutually exclusive inclusion of most of the MXEs appears tightly regulated (at least 484 of 629 MXE clusters), presenting such an example might be useful.

In case of Supplementary Fig. S29 (now S32), however, we believe that giving a visual representation of the putative ‘rescue’ we mention in the text might be helpful for the non-expert reader. In case the reviewer insists that this figure should be removed we will naturally comply.

7. Page 10: the authors should define in this section what "differentially expressed" means. As mentioned above, however, they should better use "differentially spliced", defined by the change in percent inclusion of the MXEs.

We fully agree and have made according changes throughout the revised manuscript. For details please see our answer to major concern 1.

Thank you again for submitting your work to Molecular Systems Biology. We have now heard back from the three referees who accepted to evaluate the study. As you will see, the referees are now globally supportive. We will therefore be able to accept your paper for publication pending the following modifications:

- both reviewer #1 and #2 raise remaining issues that we would kindly ask you to address with suitable amendments to the text.

In addition, from our initial pre-production checks, here are few details that we would kindly ask you to change:

-The datasets have been called out as "Supplementary Table S1-S10. They need to be called out as Dataset EV1-10.

-There is a callout for Supplementary Table S1A and B on page 21, but there is no A & B in the actual table.

-All callouts for Appendix figs have to be renamed from 'Supplementary Fig SX' -> 'Appendix Fig SX'.

-All callouts for figures are in, but the following individual panel has NOT been called out explicitly: Figure 4A

-Appendix fig S11B is called out, but the figure itself is NOT divided in A&B (but upper and lower panel). Change it to A & B?

*Appendix

- Needs to be renamed -> 'Appendix'

- Needs a Table of Content with page numbers.

- The figures should be renamed -> Appendix Fig S1-S36

- Panels in Appendix Fig S4A-C need to be specified in the legend.

-Figure 5. Tasmanian Devil (Devel) is misspelled.

REVIEWER REPORTS

Reviewer #1:

The manuscript is much improved and the authors have carefully and satisfactorily addressed most of our criticisms.

The main remaining concern is the possible confounding of the functional implications of MXEs and those of the expression levels and length of their cognate genes. For instance, by using the entire human genome annotation as background for their GO analyses, the authors are biasing their enrichments towards genes that are highly expressed and/or have more exons, in which MXEs are easier to detect. Similarly, how would the ROC curve in Figure 4D look like if it was generated based on the expression of the genes harbouring the MXEs used in the machine learner? This potential coupling needs better resolving.

Minor comment:

The increased tissue-specificity of novel MXEs (highlighted as interesting in page 12) is expected. The more tissue-specific a MXE, the less likely it is to be detected in a random sample of tissues. MXEs have been annotated based on experimental evidence from studies with unequal tissue coverage. So perhaps "interestingly" can be replaced by "expectedly".

Reviewer #2:

I think that the points raised in the previous round of review have been largely satisfactorily addressed. Some minor points should be clarified.

Minor points

In abstract section, "More than 82% of the MXE clusters are conserved in mammals, and five clusters have orthologs in *Drosophila*." The latter sentence is not clearly clarified. Authors firstly should confirm that the genes containing cluster clusters are ortholog between mammalians and *Drosophila* by phylogenetic analysis. In addition, in Fig S35, authors mention that "all three genes have an orthologous cluster of two MXEs, the MXEs have identical exon phase, similar length and sequence similarity, and code for the same region of the protein." But these parameters are not enough to determine MXE orthology. Authors might confirm exon orthology between them by phylogenetic analysis. If the exon duplicates in mammalian resemble each other more closely than any of the duplicates from *Drosophila*. This indicates that MXE are unlikely to be ancestral but have probably occurred independently in different lineages, which would be the result of convergent evolution (Trends Genet 2004, 20:171-176.). If this case, considering that MXEs have probably occurred independently in different lineages, it is not suitable to say "five clusters have orthologs in *Drosophila*".

Page 9. "and competitive RNA secondary structural elements (Graveley, 2005; Yang et al, 2012; Suyama, 2013; Lee & Rio, 2015)". In the paper (Suyama, 2013), only one pair of RNA secondary structure is found, is it competitive?

Reviewer #3:

The authors have successfully addressed my main concerns.

2nd Revision - authors' response

4 November 2017

MAX-PLANCK-GESELLSCHAFT

Editorial requests:

1. The datasets have been called out as "Supplementary Table S1-S10. They need to be called out as Dataset EV1-10.

Changed accordingly.

2. There is a callout for Supplementary Table S1A and B on page 21, but there is no A & B in the actual table.

We have changed the reference to 'Dataset EV1'.

3. All callouts for Appendix figs have to be renamed from 'Supplementary Fig SX' -> 'Appendix Fig SX'.

Changed accordingly.

4. All callouts for figures are in, but the following individual panel has NOT been called out explicitly: Figure 4A.

Figure 4A is now referenced in the main text.

5. Appendix fig S11B is called out, but the figure itself is NOT divided in A&B (but upper and lower panel). Change it to A & B?

We have labelled the panels of Appendix Fig. S11B with A and B.

*Appendix

6. Needs to be renamed -> 'Appendix'

Changed accordingly.

7. Needs a Table of Content with page numbers.

Changed accordingly.

8. The figures should be renamed -> Appendix Fig S1-S36

Changed accordingly.

9. Panels in Appendix Fig S4A-C need to be specified in the legend.

Changed accordingly.

10. Figure 5. Tasmanian Devil (Devel) is misspelled.

Changed accordingly.

Reviewer #1 (Remarks to the Author):

The manuscript is much improved and the authors have carefully and satisfactorily addressed most of our criticisms.

1. The main remaining concern is the possible confounding of the functional implications of MXEs and those of the expression levels and length of their cognate genes. For instance, by using the entire human genome annotation as background for their GO analyses, the authors are biasing their enrichments towards genes that are highly expressed and/or have more exons, in which MXEs are easier to detect. Similarly, how would the ROC curve in Figure 4D look like if it was generated based on the expression of the genes harbouring the MXEs used in the machine learner? This potential coupling needs better resolving.

The reviewer asks whether gene expression could predict with similar accuracy as exons (MXEs). This is an interesting question and we have addressed this in the revised manuscript (Fig. 4D and main text ‘Conversely, cardiac-neuromuscular disease could be predicted with an AUC of 72% using RPKM-based gene expression values (Fig. 4D)’). In a nutshell, both F1 score and AUC are decreased by ~10% when using gene expression values, which is by all means a quite drastic decrease in the predictive performance.

Of note: We have used a new R version and the most recent packages (ROCR, randomForest, caret) for the computations, which is why we (re-) analyzed the gene expression as well as the MXE data with the updated software. This resulted in small changes in the MXE predictions (slightly better AUC, otherwise almost identical, see revised Fig. 4D and main text) as compared to the original values.

The reviewer also raises the concern that using the entire human genome as background for an MXE GO enrichment analysis might skew the results because exon-rich or highly expressed genes have a higher chance to contain MXEs.

In this sole instance we humbly beg do disagree. The detection algorithm searches and validates MXEs in the entire genome. Given that the background distribution for the detection and validation algorithms is based on the whole genome, we would argue that the

selection of the entire genome for GO enrichment is adequate (statistically correct).

Please also bear in mind that the length or the multitude of exons in a gene is not a selection criterion of the algorithm. Furthermore, the GO analysis is supposed to show the enriched term for the currently validated MXEs and we cannot and would not want to make assumptions about potential 'missed' MXEs and their purely hypothetical function.

We could go on and argue that our algorithm is quite sensitive and has a good recall, that it does detect plenty of 'genes with few exons' but we will stop here for the sake of brevity.

Minor comment:

2. The increased tissue-specificity of novel MXEs (highlighted as interesting in page 12) is expected. The more tissue-specific a MXE, the less likely it is to be detected in a random sample of tissues. MXEs have been annotated based on experimental evidence from studies with unequal tissue coverage. So perhaps "interestingly" can be replaced by "expectedly".

Changed accordingly.

Reviewer #2:

I think that the points raised in the previous round of review have been largely satisfactorily addressed. Some minor points should be clarified.

Minor points

1. In abstract section, "More than 82% of the MXE clusters are conserved in mammals, and five clusters have orthologs in Drosophila." The latter sentence is not clearly clarified. Authors firstly should confirm that the genes containing cluster clusters are ortholog between mammals and Drosophila by phylogenetic analysis. In addition, in Fig S35, authors mention that "all three genes have an orthologous cluster of two MXEs, the MXEs have identical exon phase, similar length and sequence similarity, and code for the same region of the protein." But these parameters are not enough to determine MXE orthology. Authors might confirm exon orthology between them by phylogenetic analysis. If the exon duplicates in mammalian resemble each other more closely than any of the duplicates from Drosophila. This indicates that MXE are unlikely to be ancestral but have probably occurred independently in

MAX-PLANCK-GESellschaft

different lineages, which would be the result of convergent evolution (Trends Genet 2004, 20:171-176.). If this case, considering that MXEs have probably occurred independently in different lineages, it is not suitable to say "five clusters have orthologs in *Drosophila*".

We thank the reviewer for pointing this out and apologize for the ambiguous usage of the term "orthologous" here. To exclude any misunderstanding, we have changed "orthologous" to "homologous" in the manuscript and the Appendix (Appendix figures and figure legends). As we already stated in the manuscript, a thorough analysis of the potential orthology of MXEs in human and *Drosophila* requires many more species to be analyzed. A quick phylogenetic analysis showed orthology (in the sense the reviewer explained) for two MXE clusters. However, analyzing these data in enough detail for a solid statement is out of the scope of this manuscript.

2. Page 9. "and competitive RNA secondary structural elements (Graveley, 2005; Yang et al, 2012; Suyama, 2013; Lee & Rio, 2015)". In the paper (Suyama, 2013), only one pair of RNA secondary structure is found, is it competitive?

We thank the reviewer for their careful reading and removed Suyama 2013 from the references in this sentence.

Corresponding Author Name: Stefan Bonn

Journal Submitted to: MSB

Manuscript Number: MSB-17-7728